# Stenus-inspired, swift, and agile untethered insect-scale soft propulsors

Xingxing Ke[1], Haochen Yong[1], Fukang Xu[1], Han Ding[1] & Zhigang Wu ®[1] ✉

Mimicking living creatures, soft robots exhibit incomparable adaptability and various attractive new features. However, untethered insect-scale soft robots are often plagued with inferior controllability and low kinetic performance. Systematically inspired by the swift swingable abdomen, conducting canals for secretion transport, and body setae of *Stenus comma*, together with magnetic-induced fast-transformed postures, herein, we present a swift, agile untethered millimetre-scale soft propulsor propelling on water. The demonstrated propulsor, with a body length (BL) of 3.6 mm, achieved a recorded specific speed of ~201 BL/s and acceleration of ~8,372 BL/s$^2$. The comprehensive kinetic performance of this propulsor surpasses those of previous ones at similar scales by several orders. Notably, we discovered momentum-transfer-induced over-biological on-demand braking (deceleration ~−5,010 BL/s$^2$) and elucidated the underlying hydrodynamics. This work offers new insights into systematically bio-inspired artificial insect-scale soft robots, enabling them to push boundaries in performance, and potentially revolutionizing robot design, optimization, and control paradigms.

Throughout billions of years of evolution and selection, natural organisms have achieved unparalleled success in adapting to different environments on Earth through their elegant structures, ingenious implementation mechanisms, and prominent performances. This success has inspired an enormous number of researchers from diverse fields, giving birth to numerous exciting research areas and useful products. In particular, by mimicking diverse biological systems from natural living organisms, soft robotics has rapidly advanced recently, offering various features and great adaptability that open up a brand-new perspective for robotics in highly dynamic interactive scenarios[1–4]. However, compared with their swift and agile natural counterparts, soft robots often exhibit inferior kinetic performance and agility during movements/interactions[5,6]. It is particularly prominent for untethered insect-scale soft robots, where codesigning a high-performance actuation mechanism, agile control strategy, and even collaborative functionalities are severely constrained by their very limited size[7]. To address these tough technical challenges, researchers have pursued to achieve better motion controllability and kinetic performance[8]. Certainly, the landscape of small-scale robotics is adorned with a plethora

of elegant technologies that empower these diminutive marvels. Notable examples include the catalytic artificial muscle-based insect-scale robot[9], magnetic millirobot[10], jet-based microswimmers[11], Marangoni effect-based microbots[12], and a myriad of others[11]. These small-scale enabling technologies have significantly expanded the repertoire of motion forms and capabilities attainable by small-scale robots. Despite intensive efforts to develop untethered, insect-scale soft robots, their comprehensive kinematic performance (speed, acceleration, and deceleration) and on-demand motion behaviour still cannot be comparable/superior to their natural counterparts[9,13].

As a structurally simple yet effective propulsion mechanism, surface tension gradient-induced propulsion is well-suited for small-sized soft robots propelling on water surfaces in an untethered manner[14,15]. However, lacking effective control strategies for a rapid, on-demand local tension tuning ability and hydrodynamic intervention limits its enormous potential to be applicable in robot manipulation. Particularly, real-time dynamic power supply management and instant braking are rather difficult, leading to very limited manoeuvrability in dynamic scenarios. It manifests as uncontrollable

[1]State Key Laboratory of Intelligent Manufacturing Equipment and Technology, Huazhong University of Science and Technology, Wuhan 430074, China. ✉e-mail: zgwu@hust.edu.cn

instantaneous kinematics, e.g., a lack of rapid, on-demand, real-time multiple halt/start manoeuvrability, making it challenging to achieve both agile motion control and comparable comprehensive kinetic performance as that of natural insects. In nature, rove beetles have a freely swingable abdomen with a highly developed built-in pygidial gland system at their tail tip, where a liquid secretion stored in connected reservoirs, stenusin (a natural surfactant), can be delivered/released onto water on demand (Fig. 1a). Due to its elegant multiscale structural features, including a flexible swingable abdomen, a surface covered in bristles, and specialised conducting canals for fuel transport, *Stenus commas* (Coleoptera, Staphylinidae, commonly known as

rove beetles) can achieve an instantaneous high speed (400–750 mm/s or 80–150 BL/s) to dash over water surfaces. Such an agile reaction and swift movement hence enable *Stenus* to quickly escape from dangers or avoid being predated[16].

By systematically mimicking such unique multiscale biological structures, mechanisms, and motion behaviour and combining them with precise yet fast-responsive magnetic manipulation, an engineering approach and corresponding fundamental sciences are presented in this work to realise swift, agile untethered insect-scale soft propulsors (Uni-SoPros), as shown in Fig. 1b, with comprehensive consideration of the following aspects (Fig. 1c). First, a bioinspired real-

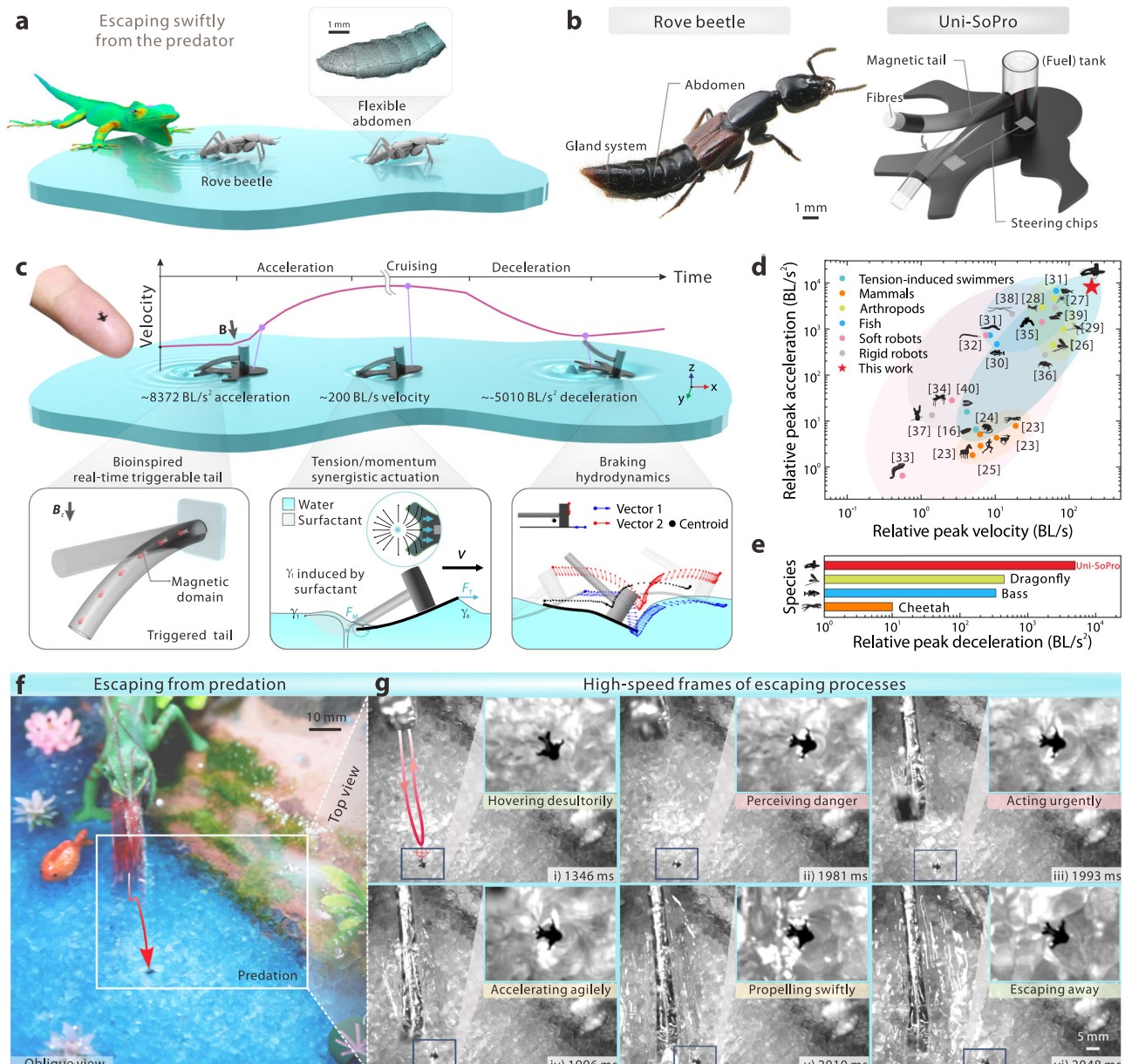

**Fig. 1 | *Stenus comma*-inspired swift and agile untethered insect-scale soft propulsors (Uni-SoPros). a** To avoid predation, a rove beetle (*Stenus comma*) can dash over the water surface at 400–750 mm/s after briefly dipping its abdomen tip and delivering secreted stenusin (a natural liquid surfactant) into water. **b** The rove beetle, has a flexible abdomen with a developed glandular system, empowering it with an on-demand release of secretion and following agile motion on water (left panel); bioinspired by the abdomen structure and on-demand trigger mechanism, an untethered insect-scale soft propulsor (Uni-SoPro) is proposed (right panel). **c** Uni-SoPro can achieve on-demand motion with high acceleration (-8,372 BL/s²),

high velocity (-201 BL/s), and high deceleration (-5,010 BL/s²), enabled by its bioinspired real-time magnetic-triggerable tail, tension and momentum synergistic actuation, and effective braking hydrodynamics. **d** Relative peak velocity and acceleration of mammals[21–23], arthropods[24–27], fish[28,29], soft robots[30–33], rigid robots[34–37], small-scale tension-induced soft swimmers[15,38] and this work. **e** Relative peak deceleration comparisons between the typical animals and Uni-SoPros. **f** By mimicking a rove beetle's behaviour, a Uni-SoPro is dodging a fast and violent predator (lizard) in a pond. **g** Key high-speed frame snapshots of the process.

time magnetic-triggered tail is introduced to on-demand, real-time transfer propulsion materials (surfactants) from a tank to water behind its caudal fin, providing primary controllability. Second, a precise release of surfactants results in a better-controlled surface tension gradient. Such a gradient difference along the contact line directly induces an expected horizontal component of the resultant, pulling Uni-SoPros forward. Simultaneously, drastic Marangoni flow impacts on the caudal fin and induces a momentum transfer to Uni-SoPros, particularly during the initial acceleration state, resulting in a synergistic actuation that guarantees swift motions. Third, by utilising fast posture transformations, a rapid momentum transfer is induced to water and attenuates the most kinetic energy, enabling Uni-SoPros agile braking in dynamic situations. Consequently, Uni-SoPros exhibits *Stenus commas*-like on-demand controllability and comparable comprehensive kinetic performance to their natural counterparts, *Stenus commas* (Supplementary Movie 1). Its comprehensive kinetic performance index (CKPI, an all-in-one benchmark defined as a product of the relative peak acceleration, $BL/s^2$, and relative peak velocity, $BL/s$, where the relative performance refers to the ratio of performance to body length) of $1.6 \times 10^6\ BL^2/s^3$ surpasses that previously reported by several orders, as shown in Fig. 1d and Supplementary Table 1. Moreover, the presented Uni-SoPro shows an agiler braking capability with a deceleration of $-5,010\ BL/s^2$ (several orders higher than its natural counterparts), as shown in Fig. 1e and Supplementary Movie 2.

Through the systematic biomimicking enabled swift movement and agile action, the presented Uni-SoPro can mimic a predation-escaping behaviour similar to its natural counterpart, as shown in Fig. 1f, Supplementary Movie 3, and some extracted snapshots in Fig. 1g. Moreover, some tough tasks in dynamic scenarios can be completed with the established trajectory programming model, e.g., passing a time-varying traffic light-controlled labyrinth, as shown in Supplementary Movie 1. These demonstrations highlight Uni-SoPro's potential not only to replicate the capabilities of biological organisms but also to surpass them in some specific points, paving a new way for systematic bio-inspiration to fully lock the potential of artificial small-scale soft robots and opening up a new window for their practical implementation.

## Results

### Systematic bio-mimicking and comprehensive codesign

Countless cycles of evolution and natural selection have given the *Stenus comma* unique mechanisms and elegant structures across scales, endowing it with superior kinematics on the water and hence avoiding predation by its natural predators. From a multi-scale perspective, these naturally optimised structures inspired us to design an artificial equivalent embodiment - Uni-SoPros, as shown in Fig. 2a. Specifically, there are three cross-scale bio-inspired structures together with artificial structures that empower Uni-SoPro. First, at the macroscopic scale, we noticed that the flexible bendable abdomen in *Stenus comma* is a critical structure that delivers propulsion materials on the water surfaces on demand. By virtue of the fast response of magnetic manipulations[17,18], a magnetic tail was introduced along the middle axis to mimic *Stenus'* flexible swung abdomen (controlled via a programmable three-dimensional Helmholtz coil manipulation platform, as shown in Supplementary Fig. 1). Such a design ensures movement along the primary axis and enables on-demand, highly dynamic manipulation of its propulsion. In cases where fuel delivery is off-centre, it would result in an asymmetric tension force distribution along the contact line, thereby inducing diagonal motion facilitated by magnetic navigation (as shown in Supplementary Fig. 2). Second, at the microscopic scale, we found that countless tiny conducting canals in the gland system serve as propulsion material/fuel - surfactant transport bridges. Inspired by this, a receiving/conducting canal-inspired surfactant subdivision transport mechanism was introduced to optimise delivery by inserting aligned fibres at the tip of the tail. Third, at the

mesoscopic scale, with bristle-filled surfaces, *Stenus comma* can prevent sinking in water, which is of significance for guaranteeing its stability during propulsion/operation on water surfaces. To mimic such a feature, micro-structured morphologies were introduced on the surface of the main body to obtain a superhydrophobic skin and hence avoid sinking. Fourth, simultaneously, to improve real-time manoeuvrability, a pair of magnetic steering chips were integrated into Uni-SoPros for real-time fast yet stable swerving control, as shown in Supplementary Figs. 3 and 4. Moreover, to decouple the propulsion and steering control, a naturally decoupled design and manipulation mode were adopted (as shown in Supplementary Fig. 5 and Supplementary Note 1). During the decoupling verification experiments, it was observed that whether the magnetic tail is bent downwards or upwards. The tail essentially does not lead to a significant deviation (<4°) from the main axis of the body when Uni-SoPro is in rotation (Supplementary Movie 4). Ultimately, such a systematic bio-inspiration and comprehensive codesign of the above aspects provide us with a potential embodiment to achieve biological-level and even over-biological motion behaviours.

To optimise the performance of Uni-SoPros and ensure their functionalities within appropriate parameters, we conducted a comprehensive investigation and characterisations of the bio-inspired structures involved, which can provide guidance for codesigning the whole system. First, we performed a series of fundamental characterisations on the magnetic particle powder and magnetised film employed (Supplementary Fig. 6). This analysis allowed us to understand the properties and behaviour of these materials, forming a crucial foundation for subsequent investigations. One of the focal points of our study was the magnetically-controlled tail. We examined the influence of the magnetic particle content and magnetic tail size scale on its behaviours of downward bending for fuel delivery and upward bending for detaching from the water surfaces, as shown in Fig. 2b–d. Our findings revealed that a magnetic tail with higher magnetic particle contents exhibited more favourable bending behaviours for ensuring effective fuel delivery to the water surfaces and on-demand cutting of the fuel supply by overcoming the solid-liquid interface energy. In addition, we also characterised the influence of different postures and submerged magnetic tail lengths on swerving behaviour (Supplementary Fig. 7). The results show that excessive immersion of the magnetic tail in water will attenuate its swerving behaviour. Thus, selecting a suitable magnetised tail with appropriate magnetic particle contents, tail size scale and submerged length of tail is very significant for achieving efficient manoeuvrability.

Regarding actuation, as mentioned earlier, maintaining a sufficient surface tension gradient is crucial for Marangoni propulsion. However, previous studies have shown that, particularly in a limited water area, surfactant molecules usually spread quickly and instantly cover the entire surrounding surface, and subsequently decrease the gradient dramatically upon initial surfactant release. Such a rapid gradient decline and consequent velocity decline leads to a narrow window for motion manipulation and further precise trajectory control/planning when propulsion materials are not precisely delivered/managed[15]. A feasible approach to hinder such a rapid transition and delay gradient decline is to precisely regulate and limit the delivered surfactant volume onto the water surface. Meanwhile, as found in previous studies, between the gland cells and reservoirs of *Stenus Comma* (Supplementary Fig. 8), synthesised secretion is gathered in a receiving structure within an extracellular cavity of gland cells, further drained by a secretion-conducting canal located in a canal cell, and transported to connected reservoirs, where a bundle of receiving and conducting canals are used to smoothly/evenly transport secretions[16]. Furthermore, to mimic such a receiving/output division, a bundle of polypropylene microfibres was introduced at the tip of magnetic tails, by roughly aligning micron-scale fibres straightly and followed twisting them into a bundle, and inserting them into artificial tail tips. In a

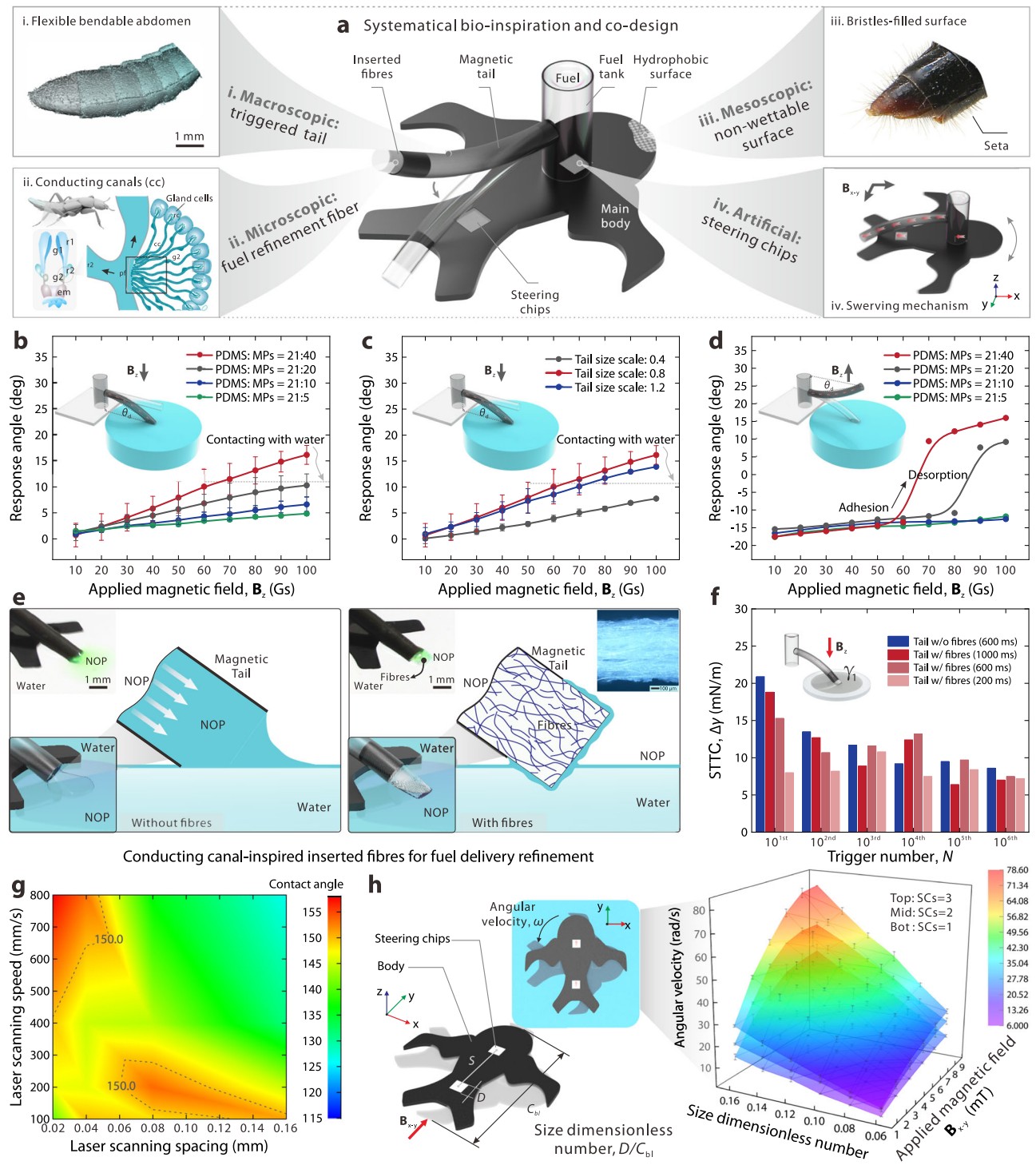

further comparison, the fuel (surfactant) was observed to dispense more homogenously with a thin layer of fuel directly contacting water from that with fibres, while, in contrast, a big drop of fuel was observed to directly contact water from that without fibres. Two fuel transport and delivery modes are illustrated in Fig. 2e. Meanwhile, we observed a gentle Marangoni flow from the delivery with fibres inserted, while strong convection from that without fibres, as shown in Supplementary Fig. 9.

By measuring the surface tension decrement after every ten triggers from the first ten times ($10^{1st}$) to the sixth ten times ($10^{6th}$) on freshwater surfaces, we studied their local surface tension tuning capabilities (STTC) with the tails with/without fibres (Fig. 2f

and Supplementary Fig. 10). There was a dramatically degressive local tension tuning capability as the trigger number increased when the fibres was not inserted. By contrast, those with fibres show a slow and linear-like attenuation of local tension tuning, particularly at a 200-ms trigger time. The attainment of a sufficient and stable STTC metric is of critical importance, as it directly impacts the overall kinematic performance and its consistency during multiple triggering. Additionally, we found that by controlling the dipping duration, it is easy to control the dipping behaviour by magnetic fields. Such precise tension tuning, which embodies energy supply regulation, offers a relatively ideal trigger attribute by providing a sufficient time window with expectable

**Fig. 2 | Systematic bio-inspiration from its natural counterparts (*Stenus comma*), and codesign of the soft propulsor with detailed characterisations.** **a** Systematic bio-inspiration designs from macro to micro as well as artificial codesign to enable the presented soft propulsor with a triggerable tail, stable fuel transport and delivery, nonwetting surfaces, and decoupled steerable capability. **b**, **c** Effects of the mixture ratio of PDMS and magnetic particles (MPs) and magnetic tail size scale on the bending response under various external applied magnetic fields $B_z$. The error bars indicate the standard deviation for $n = 3$ sample measurements at each data point. **d** Detaching behaviour of the magnetic tail from the water surface with different mixture ratios of PDMS and magnetic particles. **e**, **f** Schematic illustration of the conducting canal-inspired inserted fibres smoothing fuel delivery compared with the one without inserted fibres, and the comparison experiments of multiple local surface tension tuning show that the tail with (w/) inserted fibres shows a linear-like local surface tension tuning capability as

the accumulated trigger number increases, particularly in a short touching mode, while the local surface tension tuning capability (abbreviated as STTC) of that without (w/o) inserted fibres dramatically decreases as accumulated trigger number increases, $n = 3$ sample measurements for a referenced group (w/o inserted fibres). The error bars indicate the standard deviation for $n = 3$ sample measurements at each data point. **g** Static contact angle mapping of c-PDMS films under various laser scanning spacings and scanning speeds, among which, positions greater than 150° are suitable for being selected as nonwetting surface parameters. **h** Schematic of the rotation performance test of the soft propulsor body with different sizes and quantities of SCs under different applied magnetic fields, $B_{x-y}$ (left panel). Effect of the quantity and size of SC and applied magnetic field ($B_{x-y}$) on the swerving performance (right panel). Each point indicates the average of three sample measurements. The error bars indicate the standard deviation for $n = 3$ sample measurements at each data point.

velocity for precise manipulation, and further trajectory control/planning.

Meanwhile, we also noticed that rove beetles are covered with setae on their body and legs to prevent sinking. To mimic this feature, micro-structured morphologies were introduced by laser surface treatment on the surface of the main body to obtain super-hydrophobic skin and hence prevent sinking (Supplementary Figs. 11 and 12). The static contact angles of the c-PDMS (carbon doped PDMS) film that is used for the main body of Uni-SoPro under different laser surface treatment conditions are shown in Fig. 2g, among which, the parameter of the static contact angles larger than 150° are suitable for being selected as surface hydrophobic treatments.

Simultaneously, to improve real-time manoeuvrability, the embedded magnetic steering chips (SCs) were integrated into Uni-SoPros for real-time fast yet stable swerving control. We found that the rotation velocity of the main body increases with the number and size of SCs. In this work, to balance the steering and other motion characteristics, a pair of SCs with suitable size ($D/C_{bl} = 0.12$) were employed for the steering control. Moreover, to decouple the propulsion and steering control, a natural decoupled design and control mode were adopted, as shown in Supplementary Fig. 5 and Supplementary Note 1. Through codesigning all related parts in a compact body under the control of a solely magnetic field, Uni-SoPro achieves a swift and agile behaviour similar to that of *Stenus comma*, as shown in Supplementary Movie 5.

## Kinematic performance

By virtue of the codesign of the above systematic bio-inspiration and artificial structures, Uni-SoPro can achieve biological-level and even over-biological kinematical performance under the trigger of a single three-dimensional magnetic field (Fig. 3a). Specifically, we studied the kinetic performance of a Uni-SoPro with a characteristic size of 3.6 mm by extracting its real-time speed and acceleration (Fig. 3b). Its peak acceleration reaches ~30 m/s² or ~8,372 BL/s² after a short time (~20 ms), and the maximum kinematic speed reaches ~725 mm/s or ~202 BL/s (~250 ms later), as shown in Supplementary Movie 1. Particularly, during a braking test of a Uni-SoPro of the same size, as shown in Supplementary Movie 2, it exhibits braking with a deceleration of −5,010 BL/s², as shown in Fig. 3c, with detailed kinetic characteristics shown in Supplementary Fig. 13. During these kinematic characterisations, it is important to note that some peaks and fluctuations may appear in the acceleration curve. These variations can be attributed to potential discontinuities in the acceleration process as well as minor noise in the measurements, particularly in the presence of background light. These factors are considered possible sources of variability in the recorded acceleration data. Additionally, when an external load of 1.5 times self-weight (~12 mg) is applied, Uni-SoPro has a similar performance to that without the external load, with the peak speed reduced from ~230 mm/s to ~200 mm/s and slightly slower its acceleration, as shown in Supplementary Fig. 14.

To learn the scale effect on the kinematic characteristics of Uni-SoPros, we investigated a series of Uni-SoPros that are geometrically scaled with various characteristic sizes on movements via a high-speed recording system. The results show a clear trend where the peak velocity monotonically decreases with increasing size, particularly at smaller scales, as shown in Fig. 3d, with detailed structural parameters shown in Supplementary Table 2 and the flow mapping of each sample size propelled at its peak velocity shown in Supplementary Fig. 15. Dynamics analysis of basic propulsion states and performance trends related to characteristic lengths also agree well with the above findings, as shown in Supplementary Notes 2 and 3 and Supplementary Fig. 16. In addition, to optimise their performance at various scales, additional design modifications/optimisation might be necessary.

Since a gentler delivery strategy discussed in the previous section can significantly slow the velocity attenuation and hence expectable trajectory for longer-term propulsion, two Uni-SoPros with/without fibres inserted were compared to propel 10 one-way cycles subjected to the same magnetic triggering conditions (Fig. 3e). Due to limited space, Uni-SoPros turn around each time. We observed a rapid decline in velocity for that without fibres, whereas the Uni-SoPro with fibres demonstrated a nearly constant trend during the test window. This result suggests that the inserted fibres, which mimic the structure and mechanism of conducting canals in the beetle gland system, provide much more precise fluidic regulation/delivery. Consequently, it minimises velocity decline and simplifies later trajectory planning for multiple triggers. Moreover, a complementary numerical simulation also suggests that even released surfactant during multiple triggers can provide a more stable tension gradient for continuous propulsion (Supplementary Fig. 17 and Supplementary Note 4).

Owing to the on-demand and precise fuel release and consequently controllable start/brake, together with the on-demand swerving control, the presented Uni-SoPros show a good trajectory planning capability. As demonstrated, a Uni-SoPro with a characteristic size of 5.4 mm was preprogrammed with a variable external magnetic field **B** to precisely achieve a trajectory of a dove pattern with quite a few sharp turning points (Fig. 3f and Supplementary Movie 5), where on-demand rapid deacceleration/swerving/acceleration control is necessary.

## Braking hydrodynamics

As shown in Supplementary Movie 2, the presented Uni-SoPro exhibits an agile braking behaviour, which has been a long-standing bottleneck for tension gradient-dominated actuated artificial small-scale soft robots. To comprehend the hydrodynamics underlying and facilitating agiler manoeuvrability, we divided it into several typical states and analysed the entire braking process (Fig. 4a). During the cruising state, Uni-SoPro maintains a back-tilting posture. It sustains a high-speed cruising on the water, with a synergistic actuation of the tension gradient along the contact lines and the momentum transfer from Marangoni flow to its caudal fin. In the triggering state, subjected to the

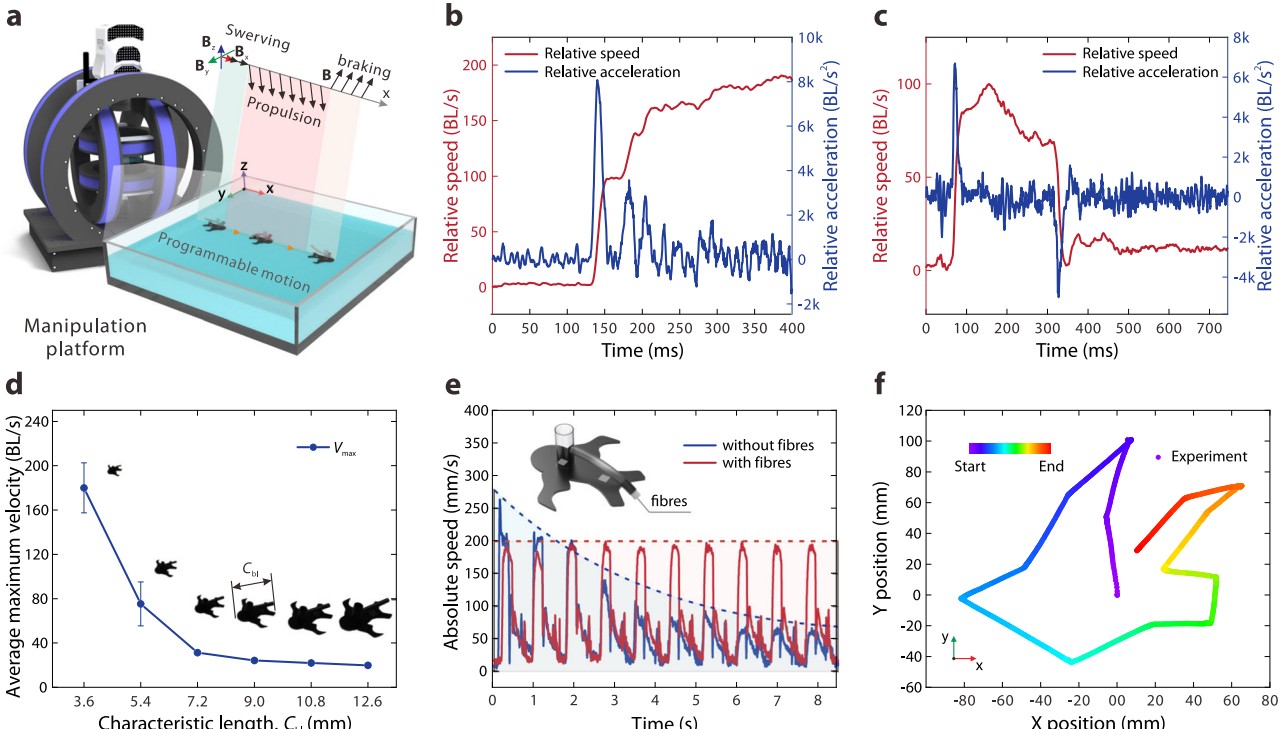

**Fig. 3 | Kinematic performance, power optimisation management, and trajectory programming. a** Schematic of the manipulation of Uni-SoPro in the three-dimensional Helmholtz coil platform. **b** The velocity and acceleration curves of a Uni-SoPro with a characteristic length of 3.6 mm in the above characterisations show a relative peak speed of ~201 BL/s and a relative peak acceleration of ~8,372 BL/s². **c** The velocity and acceleration curves of a Uni-SoPro with a characteristic length of 3.6 mm in a deceleration test (−−5,010 BL/s²). **d** Average maximum velocity of Uni-SoPros with various characteristic lengths ($C_{bl}$) in a finite water tank (200 mm × 300 mm). The error bars indicate the standard deviation for $n = 3$ sample measurements at each data point. **e** Speed attenuation comparison between a 7.2 mm Uni-SoPro with/without fibres in 10 one-way cycles. The speed of that with inserted fibres attenuates significantly slower than that without fibres. **f** Drawing a 'dove' pattern on open water by locomotion trajectory programming of a Uni-SoPro.

trigger magnetic field, the tail is cocked up and desorbs from the surface, and this motion breaks off the subsequent motive force. In the subsequent braking state, the whole Uni-SoPro tilts forward. Under the external magnetic field, such a drastic forward-tilting process induces a momentum transfer to the surrounding water and produces larger waves. This fast transformation process attenuates most of the kinetic energy of Uni-SoPro and yields agile braking. During the hitchhiking state, Uni-SoPro tilts forward to the maximum and then returns to a stable, slightly forward-tilting state. Detailed typical gesture changes during the whole process can be clearly observed in Supplementary Fig. 18.

To gain deeper insights into the braking dynamics, we conducted a series of gradient braking experiments. These experiments involved the application of magnetic fields with identical magnitudes (100 Gs), albeit with varying angles (defined as the included angles of Magnetic Field **B** and $z$-axis, denoted as $\varphi$, and here, 10°, 45°, and 60° were selected), as shown in Fig. 4b. As observed, different included angles of the magnetic fields lead to different posture changes and trajectories (Fig. 4c). By extracting key parameters (the velocity and titling angle $\beta$) of Uni-SoPros during the process, as shown in Fig. 4d, we observed that the velocity attenuation and subsequent rebounding are highly related to the external magnetic field. Such a braking behaviour can be tuned through the manipulation of magnetic fields within a small range on our current manipulation platform. In addition, we developed two equivalent speed regulation strategies, namely, intermittent propulsion and zigzag propulsion, as illustrated in Supplementary Fig. 19a. As confirmed via experimental validation, by tuning $R_i$ and $R_z$ through controlling magnetic fields, the speed can be equivalently regulated, as shown in Supplementary Fig. 19b.

Based on the experimental data and theoretical calculations, we conducted an in-depth analysis of the propelling principles and braking mechanisms and further the different forces acting on Uni-SoPros during each state during the whole process. Our investigation revealed the presence of several dominant forces, such as the momentum, hydrostatic pressure, viscous force, wave resistance, and tension that govern the overall process[19], which can be effectively described by the following hydrodynamic equation:

$$F = \int \underbrace{\rho_f g h(x) l(x) \sin\beta \, dx}_{\text{hydrostatic}} + \underbrace{\mu_f S \frac{dv}{dz} \cos\beta}_{\text{viscosity}} - \underbrace{\int_C \gamma \mathbf{t} \frac{\mathbf{v}}{|\mathbf{v}|} dl}_{\text{tension}} + \underbrace{R_W}_{\text{wave resistance}}$$
$$- \underbrace{\rho_f S_1 (v_c - v)^2 \sin\beta}_{\text{momentum}}.$$

(1)

where $\rho_f$ is the density of water, $g$ is the gravity acceleration, $h(x)$ is the depth at $x$, $l(x)$ is the length at $x$, and $\beta$ is the angle between the bottom and horizontal plane, $\mu_f$ is the viscosity of water, $S$ is the wetting area of the bottom, $dv/dz$ is the velocity gradient at the bottom surface of the Uni-SoPro, $\gamma$ is the magnitude of the tension locally, $\mathbf{t}$ is the unit vector of the direction of tension, $\mathbf{v}$ is the velocity of the Uni-SoPro, $R_w$ is the wave resistance. Particularly, our experimental analysis and calculations unveiled a new mechanism, showcasing the simultaneous collaboration of two primary driving forces (the tension gradient-induced and Marangoni flow-induced momentum transfers) during Uni-SoPro's initial acceleration phase. Such a hybrid actuation principle elucidates the underlying reasons behind the acceleration performance exhibited by Uni-SoPro. In addition, our findings indicate that the dominant force during braking (S3), as illustrated in Fig. 4e and elaborated in

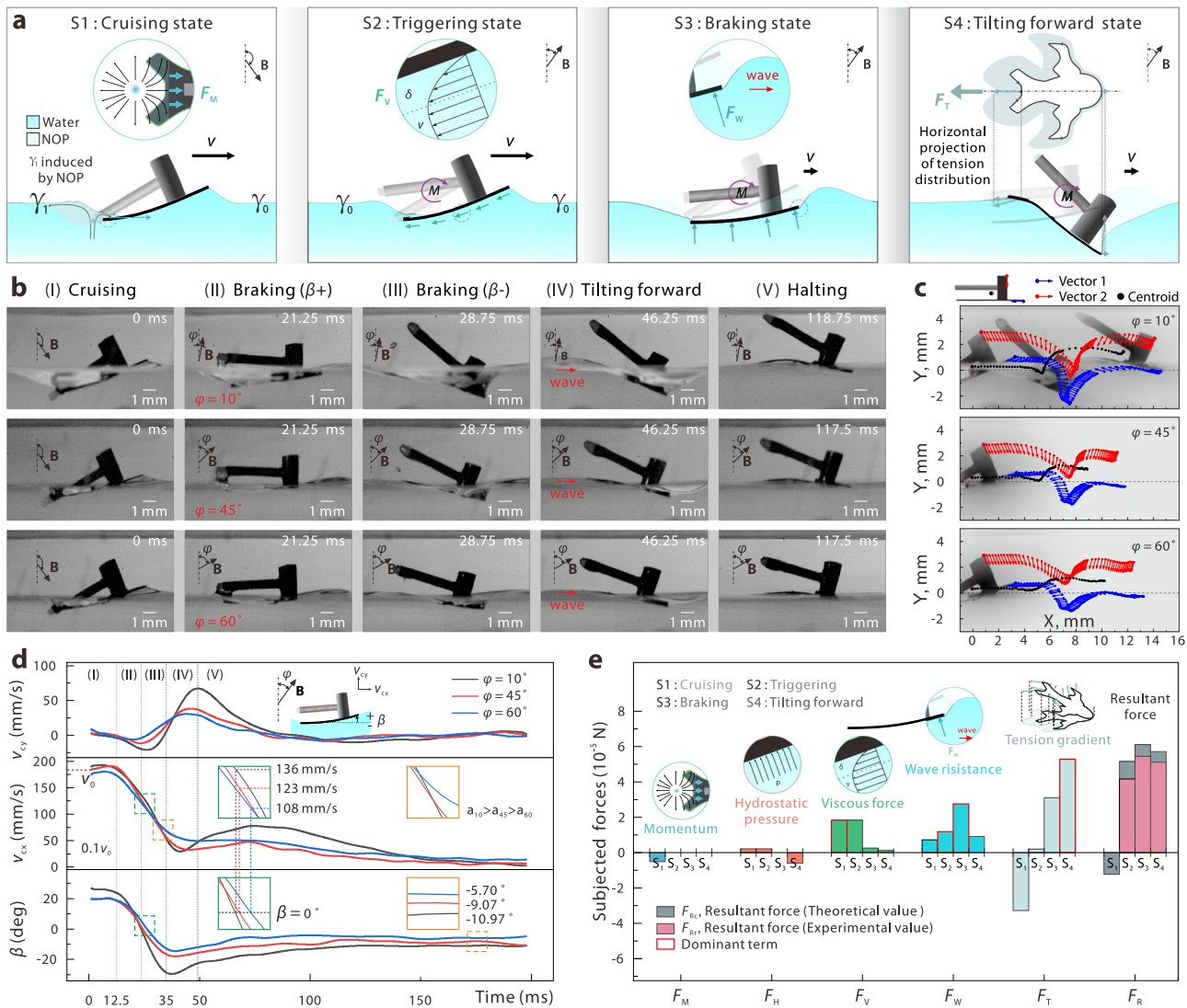

**Fig. 4 | Braking hydrodynamics analysis. a** Schematic diagram of the braking process: (S1) Steady cruising state under the actuation of surface tension gradient and momentum transfer of Marangoni flow to the caudal fin of Uni-SoPro. (S2) In the triggering state, the magnetic tail is cocked up by the external magnetic field to terminate fuel supply and the motive force. (S3) Immediately, subjected to the magnetic torque from the magnetic tail and steering chips, the induced fast transformation of Uni-SoPro from the tilt-back state to tilt-forward state triggers a drastic momentum transfer to the water, where this momentum transfer greatly attenuates the kinetic energy of Uni-SoPro. (S4) Uni-SoPro tilts forward to a maximum state and subsequently returns to a stable state under the combined action of the magnetic torque and surface tension. **b** Experimental verification of the whole braking process under different braking magnetic fields. **c** Full cycle record of the position and attitude of Uni-SoPro in the above experiments. The time interval between the sampling points is 1.25 ms in the braking process and 6.25 ms in the cruising process. **d** Full cycle record of the velocity and attitude of Uni-SoPros, and the results show a high relevance between the velocity and attitude. **e** The changing of the subjected forces in different states (from State S1 to State S4).

Supplementary Note 2, is the momentum transfer-induced force (wave resistance). This critical hydrodynamic mechanism underlies a rapid posture transformation-induced braking of Uni-SoPros.

Targeting a deeper understanding of the hydrodynamics at play during Uni-SoPro's agile braking, we utilised a six-degree-of-freedom model with a characteristic size of 7.2 mm to qualitatively illustrate the entire process via computational fluid dynamic analysis (Fig. 5). It was triggered to accelerate to a similar state with an experimental-level speed (~200 mm/s in Fig. 3d) and acceleration. Subsequently, it was triggered under magnetic torques to tilt forward and break off the power to achieve braking, with the detailed boundary conditions and loads provided in Supplementary Note 5. The full-circle kinetic/gravitational energy of Uni-SoPro, surface/kinetic/gravitational energy of water, horizontal drag force, horizontal velocity, and posture of Uni-SoPro were calculated throughout the entire process. The results indicate that the horizontal velocity attenuates within approximately

15 ms, resulting in the rapid change in the tilting angle during the braking state. From the back-tilting state to the horizontal state, ~92.5% of the kinetic energy drastically attenuates and transfers to the kinetic energy of water and the gravity gravitational energy of Uni-SoPro, as shown in Supplementary Movie 6, which agrees well with the findings from experiments.

## Kinematic modelling for trajectory programming

The absence of an effective braking mechanism in biological escape behaviours observed in *Comma stenus* and tension gradient-dominated actuation of artificial small-scale soft robots hinders the ability to achieve swift and agile multiple movements. Such a systematically bio-inspired design of Uni-SoPros successfully overcomes such a dilemma, particularly in multi-segment swift and agile propulsion scenarios. Specifically, we developed a kinematic model to program the movement of Uni-SoPro. This involved decomposing the multiple

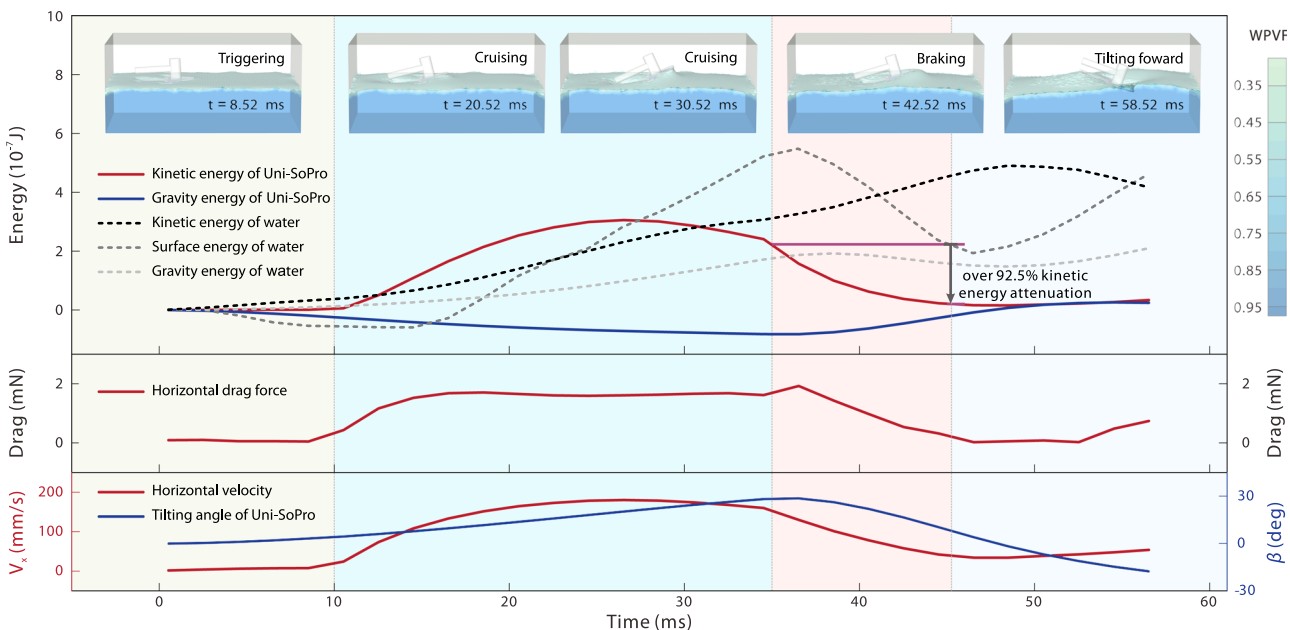

**Fig. 5 | Computational fluid dynamics (CFD) analysis of the whole braking process.** A 6-DOF model was employed to analyse the braking fluid dynamics. The full-circle kinetic/gravitational energy of Uni-SoPro and the surface/kinetic /gravitational energy of water were analysed in detail. The results verify that ~92.5% of the kinetic energy was attenuated during the tilting attitude-changing period. WPVF indicates the water phase volume fraction.

discrete movements into subsegments (Fig. 6a). Each subsegment can be further divided into four stages: the acceleration, stable propulsion, deceleration, and in-situ swerving. The typical velocity profile of each segment is illustrated in Fig. 6b and can be expressed as follows:

$$
\begin{Bmatrix} \theta \\ v_{f1}(t) \\ v_{f2}(t) \\ v_{f3}(t) \\ v_{f4}(t) \end{Bmatrix} = \begin{Bmatrix} \theta_n & t_0 \le t, < t_4 \\ a_1 - e^{b_1(t-t_0)+\ln(a_1)} & t_0 \le t < t_1 \\ v_{f1}(t_1) \cdot e^{b_2(t-t_1)} & , & t_1 \le t < t_2 \\ v_{f2}(t_2) \cdot e^{b_3(t-t_2)} & t_2 \le t < t_3 \\ v_{f3}(t_3) \cdot e^{b_4(t-t_3)} & t_3 \le t < t_4 \end{Bmatrix} \quad (2)
$$

where $\theta_n$ represents the direction of the nth segment trajectory. The velocity functions $v_{f1}(t)$, $v_{f2}(t)$, $v_{f3}(t)$, and $v_{f4}(t)$ correspond to the acceleration process, stable propulsion stage, the decelerating stage and the stationary swerving stage, respectively. The parameter $a_1$ determines the peak velocity, while $b_1$ - $b_4$ determines the control of the rate of acceleration and attenuation of speed. By incorporating the effect of surfactant release attenuation on speed and considering the fixed values of $T_{up}$, $T_d$, and $T_s$ for the specific Uni-SoPro and magnetic trigger condition, the control model can be further derived as:

$$
\begin{cases} \theta = \theta_n \\ v_f(n,t,T_t) = k(n) \cdot v_f(t,T_t) \\ l_n = \int v_f(n,t,T_t)dt \\ k(n) = \begin{cases} e^{-\lambda(n-1)}, (\text{w/o fibres}) \\ \kappa(n-1)+1, (\text{w/fibres}) \end{cases} \end{cases} \quad (3)
$$

Among them, $\lambda$ and $\kappa$ are the correction attenuation coefficients without/with fibres, respectively. In particular, when the fibres is inserted, $\kappa$ is approximately equal to zero in fewer trigger cycles. By utilising this control model, we can perform an inverse solution to determine the critical parameter $T_t$ of the nth segment trajectory. A more detailed description of the modelling process can be found in Supplementary Note 6.

The established kinematic model was further validated through experiments conducted on Uni-SoPros, both with and without inserted fibres. Remarkably, the predictions of the proposed model were found to be consistent with the experimental results in both cases, where Uni-SoPro with fibres shows an obviously more stable velocity and smaller errors than that without fibres in the multiple movement test (Fig. 6c, d). Such a kinematic performance, as well as precise trajectory programming, makes it promising to tackle some challenging scenarios.

**Swift and agile propulsion in dynamic scenarios**
Through a systematic bio-inspiration and codesign of the structures, actuation, interface, propulsion material regulation, decoupling control strategies, and hydrodynamics in such a tiny robot, Uni-SoPro can be used to unlock the potential of insect-scale robots to biological-level performance, e.g., a 3.6 mm-sized Uni-SoPro demonstrated the ability to evade the predation of a lizard with a fast and violent protruding tongue measuring 8 mm in width and up to 70 mm in length, as already shown in Fig. 1f and g. Especially, in such a scenario, the lizard's tongue strikes with an average elongation speed of 1,945 mm/s, completing the striking process within 27.5 ms. Such an escaping action showcases a kinematic performance comparable to its biological counterpart. However, for robots working in unstructured environments, agile controllability is crucial. By virtue of decoupled magnetic control, our Uni-SoPro can quickly propel/halt/steer at any time in dynamic scenarios and achieve continuous high-performance propulsion simultaneously.

To demonstrate its programmable motions, we set a labyrinth outfitted with preprogrammed time-varying signal lights that periodically change their states every 2 s (Fig. 7a). We stipulated that Uni-SoPro should not pass through the light once illuminated. When the light is off, the Uni-SoPro requires passing through the labyrinth as quickly as possible without hitting the walls from the start point to the endpoint in an open-loop control manner without any feedback mechanisms. Such a setup makes precise/instantaneous halts necessary when the light is on and instantaneous starts necessary when the light is off. Based on the above spatiotemporal constraints, the accumulated stroke error $\Delta u_{overall}$ should be within 14 mm, and $\Delta u_{1,6,16}$ should be smaller than −5.5 mm during the

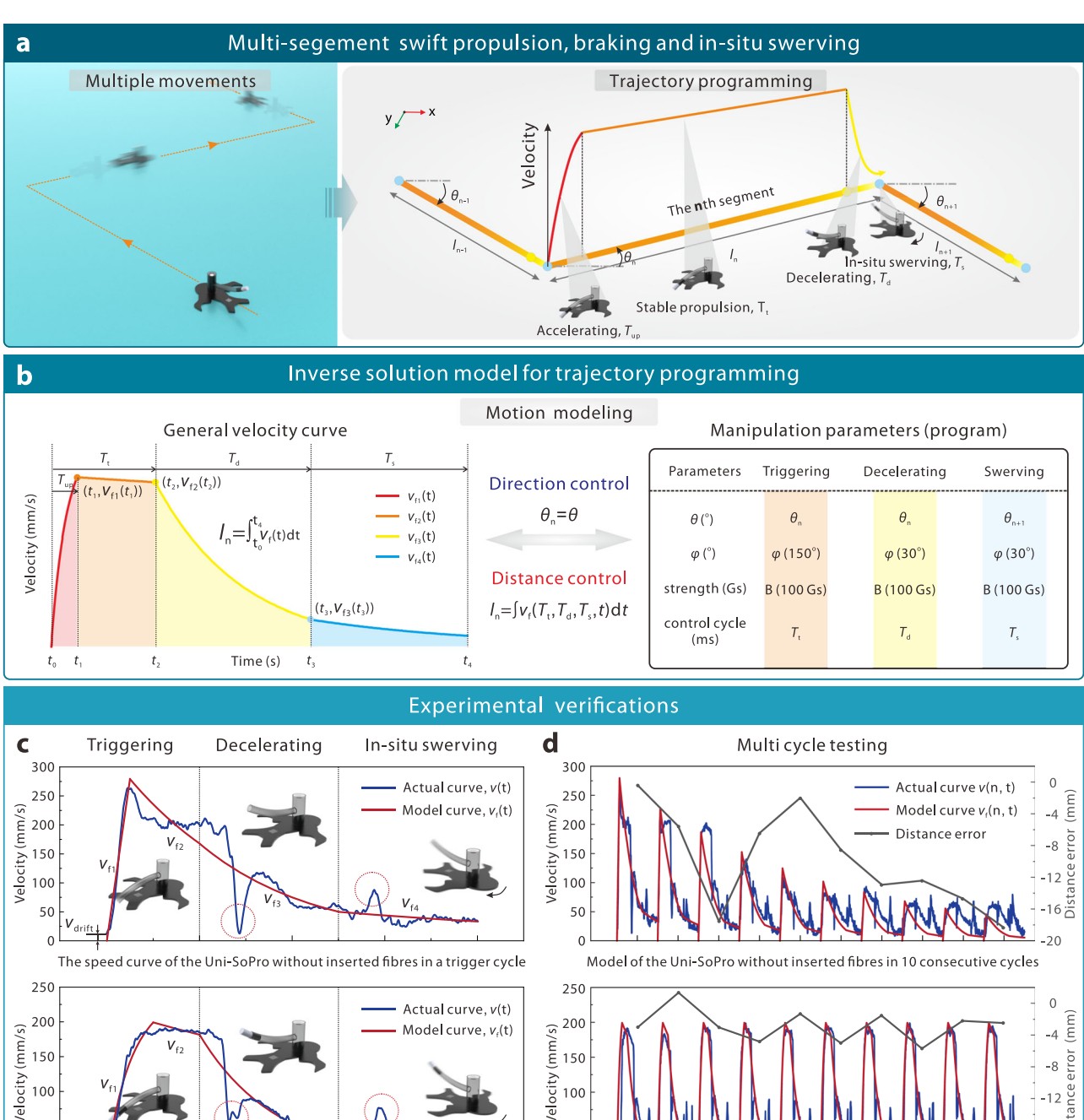

**Fig. 6 | Kinematic modelling for multi-segment motion trajectory planning.**
**a** Multiple movement programming can be further decomposed into corresponding subsegments. Each subsegment can be divided into four typical stages: the acceleration, stable propulsion, deceleration and in-situ swerving. **b** According to the above typical four motion stages, a general motion model was proposed for whole movement programming. With the established inverse solution model, the critical manipulation parameters of each subsegment can be reversely solved. **c** Experimental and model velocity comparison of Uni-SoPros ( ~ 7.2 mm) with/without fibres inserted. **d** Ten consecutive cycle kinematic curve comparisons of the predicted and experimental curves and the corresponding error analysis.

whole passing process (Supplementary Fig. 20 and Supplementary Note 7). The demonstrated Uni-SoPro only has ~0.25 s to halt for each turning, or else it will miss the best-turning window. Moreover, compared to that in open water, e.g., in Fig. 3f, there are some disturbances due to the influence of the wall on the tension distribution and the meniscus along the walls, and these nonlinear boundary conditions hence make precise trajectory control difficult. Therefore, there are quite a few stringent constraints for Uni-

SoPro to pass through such a dynamic scenario without any feedbacks (Fig. 7b). Despite those challenges, the demonstrated Uni-SoPro successfully halted and crossed each traffic signal, turned at intersection without any wall strikes during the process, and finally reached the endpoint within 20 s in a preprogrammed manner, as shown in Fig. 7c and Supplementary Movie 7, with the key snapshots of braking, propelling, and steering illustrated in Fig. 7d and e.

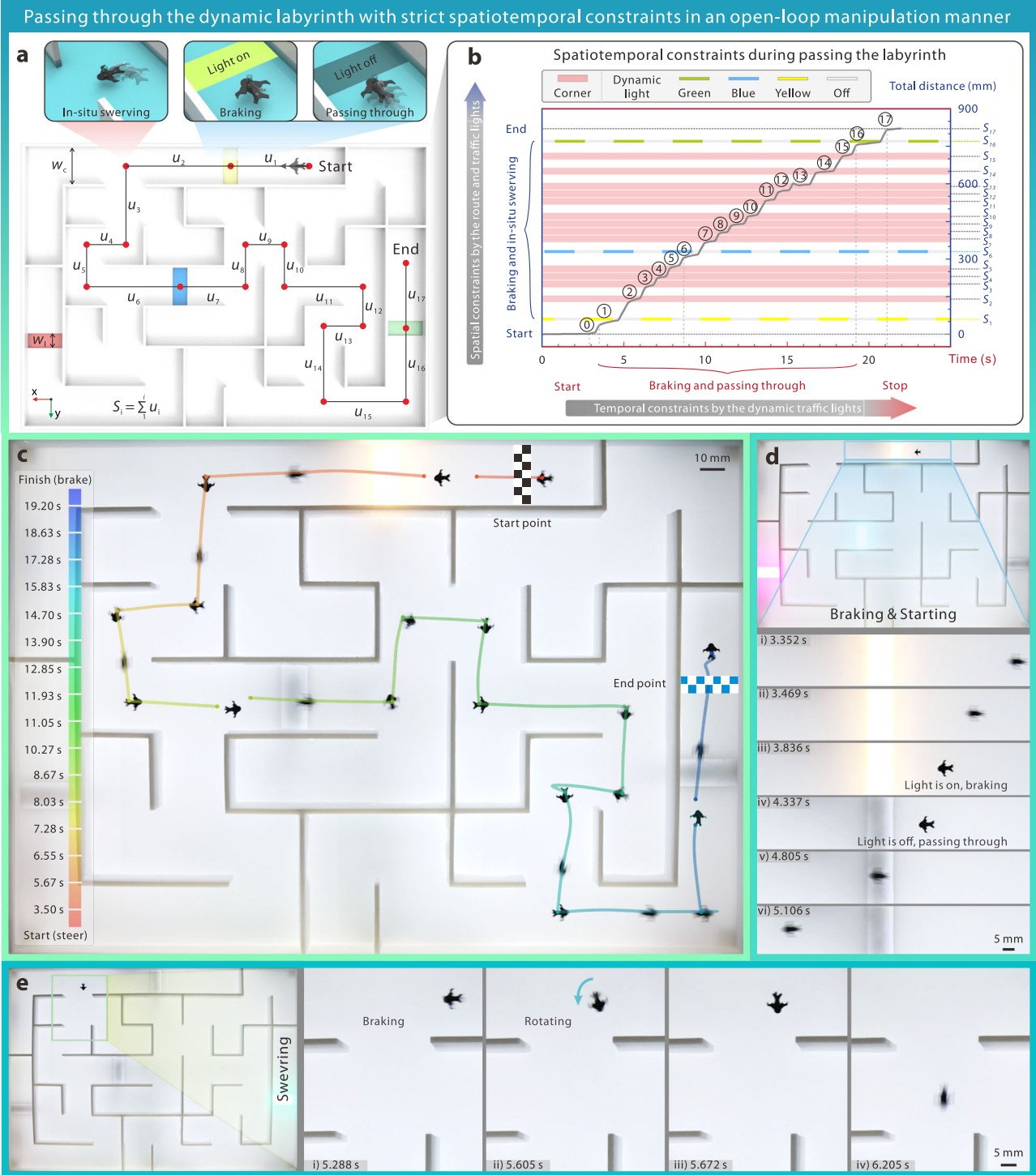

**Fig. 7 | Demonstrations of Uni-SoPro going through a dynamic labyrinth with complex spatiotemporal constraints. a** To demonstrate its controllability, Uni-SoPro should pass through such a 7×10 water labyrinth as soon as possible, following the following defined rules: a) In-situ stationary swerving at each intersection/corner. b) Obeying the signal traffic light: it can only pass through when the frontal signal light is off and should brake in front of the light when it is on. **b** Strict spatial and temporal constraints during passing through such a dynamic labyrinth. **c** A Uni-SoPro crosses through a labyrinth with time-varying signal lights. **d** Close-up snapshots of Uni-SoPro when it brakes and goes through one of the signal lights. **e** Close-up snapshots of Uni-SoPro when it steers at turning.

## Discussion

A systematic bio-inspired approach from the macro architecture to micro details and engineering embodiment of untethered, insect-scale soft propulsors introduced here empowers soft robots with over-biological on-demand motion and kinetic performance. Specifically, elegant biological structures and unique behaviours urge us to deeply understand the fundamental principles, functional mechanisms, and

action patterns behind insects, and in return enable our systematic-level engineering embodiment of Uni-SoPro, and even further motivate us to perform technical modelling to assure well-controlled and reveal extraordinary braking kinematics. The key findings are as follows: (i) a *Stenus'* abdomen and gland system inspired bionic secretion (surfactant) delivery mechanism enables on-demand local tension tuning and thus empowers Uni-SoPro with real-time power

management capability; (ii) a tension gradient and Marangoni flow-induced momentum transfer hybrid actuation principle was introduced, enabling Uni-SoPro to reach a CKPI of $1.6 \times 10^6$ BL$^2$/s$^3$, and according to the literature survey, it surpasses current state-of-the-art performance (with a CKPI ~ $6.1 \times 10^4$ BL$^2$/s$^3$) by several orders at the same scale and is comparable to fast-moving insects in similar moving modes, e.g., *Stenus*, and even superior to that of some other fast-moving natural livings moving on water, e.g., water skis (with a CKPI, ~$8.4 \times 10^4$ BL$^2$/s$^3$); and (iii) a posture-induced momentum transfer hydrodynamic braking mechanism was discovered and enables Uni-SoPro with previously inaccessible agility, particularly on barking (~ −5,010 BL/s$^2$), as shown in Supplementary Movie 8. All these features affirm a sufficient suite for achieving artificial, insect-like propulsor, and further encourage robot communities to deeply exploit unique biological features and fuse them into insect-scale robot design towards extraordinary real-time controllability and kinetic performance for highly interactive scenarios.

There are also a few limitations in the current study. A significant limitation is that during the manipulation of Uni-SoPro in this work, an open-loop grogram is implemented without real-time sensing and feedback. Thus, it does not allow for dynamic adaptation and adjustment based on the surrounding environment or changing conditions. In future studies, it would be valuable to explore the integration of a global vision system, or preferably but challengingly an energy-efficient local sensing system with edge computation to enable closed-loop manipulation of the Uni-SoPro. Additionally, the absence of a selective manipulation mechanism limits their more complex swarm behaviours. In future research, it would be beneficial to investigate and develop elegant selective manipulation strategies that can enable the robots to realise sophisticated collective behaviours and interactions.

The development of soft robots is in a bottleneck period when the performance and controllability of soft robots are not comparable with biological livings. Bio-hybrid and bio-inspired robots and high-performance soft robots are proposed as grand challenges and desirable research directions[8,20]. From such a perspective, we think this work is of significance in addressing these challenges in the current development stage and provides a good bioinspired paradigm for future robot design to fully unlock the potential of soft robots. This work also hints that through systematic exploration of these elegantly evolved biological structures and principles, we can uncover novel design concepts and enable biological or over-biological artificial robots. We can gain a deeper understanding and interpretation by delving into these principles and the science underneath such phenomena. Indeed, the idea and principle of the on-demand fuel delivery and momentum transfer braking mechanism can be applied or translated to various other similar water-skimming robots. The potential of achieving these outcomes is not confined to soft materials alone; it can also be realised by employing well-designed rigid, jointed mechanisms as well as soft and flexible mechanisms. The key to making such a system work is the introduction of triggerable smart artificial muscles for power management and momentum transfer-based braking mechanisms. We aspire for our research to set a new paradigm of systematic bio-inspiration for robotic designs. We firmly believe that through further exploration of various ingenious designs inspired by nature and the harnessing of their intrinsic advantages, more elegant multiscale structures will emerge, ultimately pushing the boundaries of the field of robotics.

## Methods
### Micro-CT image reconstruction
The structural details of the rove beetle abdomen were obtained by scanning a preserved rove beetle specimen (*Xantholinus linearis*) using microcomputed tomography (micro-CT, viva CT 40, Switzerland), segmented and three-dimensionally reconstructed by an imaging system (Mimics Research 14.0).

### Material preparation
Uni-SoPro was made of two main materials, c-PDMS and m-PDMS. They are silicone base of PDMS (Sylgard 184, Dow Corning Corp.) blended with carbon black (XC72R, Cabot) at a weight ratio of 10:1:0.5 (silicone base: curing agent: carbon black), or so-called c-PDMS; and micro magnetic particles (NdFeB, MQFB-B-20076-089, Magnequench) at a weight ratio of 20:1:40 (silicone base: curing agent: NdFeB particles), or so-called m-PDMS. The above two mixtures were prepared by a digital stirrer (RW 20, IKA) at 2,000 rpm for 3 min and then vacuumed for 5 min to remove bubbles. A surfactant, 1-octyl-2-pyrrolidone (NOP, 98% mass fraction, Macklin), was selected as the fuel without further treatment. It is a very powerful surfactant according to our comparison experiments in the previous work[15]. Other alternative surfactants can also be employed according to practical demands.

### Fabrication processes
Uni-SoPro is composed of four main parts, namely the main body, fuel tank, magnetic tail, and a pair of steering chips (SCs). The main body and fuel tank were made of c-PDMS, while the tail and SCs were made of m-PDMS (Fig. 1b). The entire process is illustrated in Supplementary Fig. 12 with an elaborate description in the additional material and methods in the supplementary information. The obtained Uni-SoPro was magnetised under an ~2.4 T pulsed magnetic field (MAG-3000, CH-Magnetoelectricity Technology Co., Ltd). To optimise its performance, we inserted a bunch of polypropylene (PP) fibres into the tube at the end of the magnetic tail. We made a series of Uni-SoPros with different size scales from 0.4 to 1.4. The detailed design parameters are shown in Supplementary Table 2.

### Tail deflection characterisation
The response of the magnetic tail with varying parameters was characterised by measuring the deflection angle $\theta_d$ under magnetic field strengths ranging from 10 to 100 Gs. Specifically, the magnetic tails (size scale of 0.8) containing various ferromagnetic particles concentrations (mass ratios PDMS to ferromagnetic particles at 21:5, 21:10, 21:20, and 21:40, respectively) were chosen to investigate the effects of ferromagnetic particle concentration on their magnetic response behaviours in terms of deflection, dipping in water, and detaching from water. Additionally, magnetic tails (with a mass ratio of PDMS to ferromagnetic particles at 21:40) of different scales, ranging from 0.4 to 1.2, were selected to characterised their downward deflection characteristics.

### Swerving investigation
SCs of various sized and numbers were embedded into the main body to learn their swerving capability (angular velocity and oscillation) under various magnetic strengths. Specifically, we selected the main body embedded with SCs with size dimensionless numbers of 0.06–0.16 and three embedded types. The process was recorded by a high-speed camera (R1212, Phantom) at 2,000 fps under the illumination of a white-light source (SOLA AM 5-LCR-VA, Lumencor) with a trigger from an external rotating horizontal magnetic field $\mathbf{B}_{x\text{-}y}$ that varied from 1 mT to 8.5 mT in steps of 1.5 mT. The angular velocity $\omega$ and oscillation stability time $t$ were extracted by imaging processing. The algorithm for obtaining $t$ was as follows: a few keyframes were picked with appropriate intervals, and the angle $\theta_{lc}$ was measured through the camera image processing program (PCC 3.6) and fitted with the following underdamped vibration equation: $\theta_{lc}(t) = Ae^{-\beta_s t}\sin(\omega_f t + \varphi_s) + C_s$, where $A$, $\beta_s$, $\omega_f$, $\varphi_s$, and $C_s$ are the fitted parameters. We used the angular velocity $\omega_f$, which equals the average angular velocity from the initial position to the equilibrium position for the first time, and the oscillation stability time $t$, which equals the feature time $1/\beta_s$ of the damped oscillation, to represent the response speed and the stability characteristics of the main body under various field strengths (Supplementary Fig. 4), respectively.

## Surface tension measurements

To learn the effect of dispersed surfactant (NOP) on water surfaces, we measured the decrement of the surface tension induced by the constructed fluid storage and delivery/release system, where the magnetic tail and fuel tank was with a size scale of 1.4. A bundle of aligned polypropylene (PP) fibres was inserted at the tip of the magnetic tail for the comparison experiment (length of ~4 mm and weight of ~0.1 mg). The magnetic tail with fibres continuously touched (interval 500 ms) the water in a Petri dish (diameter of 57 mm) for 200 ms, 600 ms, and 1,000 ms to deliver the surfactant under the trigger of a 10 mT magnetic field, $B_z$. In a reference group, the same magnetic tail without fibres was touched with the water for 600 ms at the same trigger condition. The surface tension of the water in the Petri dish after the $10^{1st}$, $10^{2nd}$, $10^{3rd}$, $10^{4th}$, $10^{5th}$, and $10^{6th}$ triggers was measured by a surface tensiometer (BZY-2, CNSHP). The Petri dishes and water were renewed after each trigger test (the interval was 60 s). The surface tension decrement of each trigger test was compared with the initial state of water (72.6 mN/m, 20 °C). A linear and an exponential function were employed to fit the data of the sample with/without fibres, respectively (Fig. 2f). The corresponding characterisation magnetic field is shown in Supplementary Table 3.

## Kinematic characterisations

All kinematic characterisations were carried out in a rectangular plastic container (18 cm × 28 cm × 3.5 cm), which was filled with deionized water (20 °C, Milli-Q Reference, MilliporeSigama) of ~2 cm depth. The container was placed in the centre of a 3D Helmholtz coil (3CHY20–100, CH-Magnetoelectricity Technology Co., Ltd) that generated a magnetic field controlled by a computer program or manual handle. The same high-speed camera at 2000 fps was used to record the kinematic motions (Figs. 1f, g, 2h, 3b–e, 4b–d, and 6 and Supplementary Figs. 2–5, 7, 9, 13 and 14, 16, 18 and 19). The speed and acceleration of Uni-SoPros were then extracted through the obtained videos within MATLAB R2020b. In Fig. 3d, a series of Uni-SoPros with size scales from 0.4 to 1.4 were measured at their relative peak speed (absolute speed divided by body length, BL/s) in a nearly linear motion. The characterisation magnetic field is shown in Supplementary Table 3. Each sized Uni-SoPro was prepared with three samples to calculate the average velocity and error bar. Among them, Uni-SoPros with a size scale of 0.4 was selected to measure its real-time speed, acceleration and deceleration (Fig. 3b, c). A size scale of 0.8 was selected to measure its speed during five reciprocating motions, and the speed attenuation behaviours of Uni-SoPros with/without inserted PP fibres were observed (Fig. 3e). For the above kinematic performance characterisations, the high-speed camera was equipped with the AF-S Micro 60 mm f/2.8 G ED lens and Sigma Macro 105 mm f/2.8 DG OS HSM. There was no extra magnification applied, and each actual length of the Uni-SoPros was calibrated using the scale within the camera's image processing program (PCC 3.6). Considering the influence of optical noise, based on our verification, there will be ~ 0.1 mm position error (the characteristic lengths of Uni-SoPros are from 3.6 mm to 12.6 mm, and the positional error is less than 5% compared with their characteristic lengths). In addition, for trajectory control, a 2D pattern of a dove pattern was designed in AutoCAD 2020, transformed into the program for the Helmholtz coil control program and executed by a Uni-SoPro with a size scale of 0.8. The SLR camera at 120 fps was used to record its position and then obtain its trajectory via MATLAB. For agile moving, to demonstrate the agility of Uni-SoPro, a size scale of 0.8 was selected to move in a continuously turning magnetic field under the SLR camera at 120 fps (Supplementary Movie 2).

## Experimental braking analysis

Uni-SoPros with a size scale of 0.8 were used to analyse the braking process. A high-speed camera was used to record the postures in a horizontal view. Since the tip of the main body and the top of the tank are relatively small deformation parts, we defined two vectors (Vector 1 and Vector 2) to mark its position and posture in the whole process. Simultaneously, according to these two vectors, we calculated the centroid of Uni-SoPro. Based on these vectors and centroid, the velocity and posture (tilting angle) of Uni-SoPro in the braking experiments were recorded and quantitatively analysed accordingly.

## Computational fluid dynamics analysis

To comprehend and illustrate the braking mechanisms of Uni-SoPros, the commercial software Ansys Fluent (ANSYS, Inc., USA) is utilised for computational fluid dynamics (CFD) simulations to quantitatively analyse its hydrodynamics with a specifically constructed model. By utilising a CFD method, we calculated its hydrodynamic drag and energy transfer during the whole moving process. Specifically, a six-degree-of-freedom (6-DOF) model was employed to compute and analyse the kinetic characterisations of Uni-SoPro. In this model, the magnetically/hydrodynamically induced deformation of Uni-SoPro was ignored, the whole body was considered rigid. However, the elastic effect was still considered in the load conditions. The mass, moment of inertia, magnetic moment, and Marangoni force in each motion state were defined in the user-defined functions (UDF). The commercial post-processing software Tecplot 360 (Tecplot, Inc., USA) was utilised for statistical analysis and visualisation of the data. Detailed settings are shown in Supplementary Note 6.

## Finite element analysis

To model the Marangoni effect upon fuel delivery, the fluid interface feature of the free surface (the surface tension was defined as a function of the surfactant concentration) was added to a laminar flow whose initial velocity field was defined as zero. Free tetrahedral elements were employed using a commercial program (COMSOL Multiphysics 5.6) with customised moving meshes to ensure computational accuracy and convergence. Simultaneously, a two-dimensional FEA liquid-solid coupling model was constructed to simulate the flow field of each sized Uni-SoPro at its peak speed. The detailed modelling process and results are shown in Supplementary Fig. 17 and Supplementary Note 4.

## Brief overview of kinematic performance

To compare the kinetic performance fairly, their peak relative speed, peak relative acceleration, and comprehensive kinematic performance index of the soft propulsor with various animals and robots were calculated, as shown in Supplementary Table 1. In addition, the detailed setups and supporting designs of various scenarios for Uni-SoPro propulsion and demonstrations and other relevant information are depicted in Supplementary Methods.

# Data availability

The authors declare that the main data supporting the findings of this study are available within the article and its Supplementary Information files. Extra data are available from the corresponding author upon request.

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

## Acknowledgements

This work was supported by the National Natural Science Foundation of China (52188102). We sincerely thank Jianing Wu (SYSU) for providing rove beetles for our study on their biological structures and behaviours; Chunze Yan (HUST) for help with the three-dimensional micro-CT reconfiguration of the rove beetles; and Cheng Zhou (THU) for discussions on vibrating sample magnetometer characterisations for soft magnetic materials.

## Author contributions

X.K., H.Y., and Z.W. conceived this project. X.K., H.Y., and F.X. carried out the experiments and characterisations. X.K. and H.Y. analysed the data and developed the models. X.K. performed the simulation. X.K., Z.W., and H.D. drafted this paper. All authors participated in the discussions of the research.

## Competing interests

The authors declare no competing interests.

## Additional information

**Peer review information** : *Nature Communications* thanks Massimo Mastrangeli, and the other, anonymous, reviewer(s) for their contribution to the peer review of this work. A peer review file is available.

