## [Peer Review File · Nature Communications]

REVIEWER COMMENTS

Reviewer #1 (Remarks to the Author):

The manuscript describes a novel unthetered bio-inspired robot of millimetric size capable, under external magnetic control, of maneuvers over the surface of water of unprecedented dexterity and performance. While the propulsion method is not novel, as it is based on a classic Marangoni effect, the device is innovative for a set of elements: unprecedented speed and acceleration, which competes and even surpass that of the original biological inspiration; controlled delivery of the propellant by magnetically-controlled trigger as well as bio-inspired surfactant delivery mediated by fibers; superhydrophobic body, which prevents the robot from being submerged and as importantly allows the braking maneuver; the braking maneuver itself, which involves magnetic control of the pose of the robot and transfer of its kinetic energy to the surrounding water. As a result, the robot is capable of navigating structured and unstructured water domains at unprecedented speed and with high degree of control and precision under open-loop programmed control. The manuscript is supported by extensive supplementary material including a substantial supplementary text and very expressive videos (which also provide frames included in some of the illustrations).

The work is substantiated by a commendable amount of work - including analytical and FEM modeling, material characterization, high-speed experimental tracking - that elucidates relevant details in most of the aspects of the research. The work stands out in the field of small-scale robotics as ingenious, innovative, and of high quality overall. We support its publication provided the points raised below are addressed.

The quality of the work is not reflected in the quality of the writing. The main text could be more concise in several places, in particular it makes frequent claims on the importance of bio-inspiration and other general comments, which are legitimate but render the text verbose, and this should be avoided. Importantly, English phrasing and grammar and the use of some words (such as "livings", "power(ing) materials") needs a thorough revision throughout the text.

By way of text formatting, most of the symbols in eq. 1 to 3 are not defined in the text (and only some in the supporting text). By the same token, the explicit reference "the previous Science Robotics perspective and viewpoint articles" in line 352 should be removed.

What software was used for the CFD analysis? This is not reported in page 11.

In Figure 16f the exponential fitting for the case of the tail with fiber 600 ms does not seem accurate and fair, in fact the STTC for the 5th and 6th set seem comparable and higher than for the fiber cases.

In line 122-123 and figure 2D the authors show that a lower magnetic particle concentration in m-PDMS improves the performance of magnetic control. We wonder what is the reason for this, is that because of the lower stiffness of the magnetic tail for lower particle density? Did the author detect a lower limit of particle density in the tail that still makes the robot functional?

The authors claim that the horizontal (swerve, pose) and vertical (trigger, propulsion) control of the robot can be and in fact are decoupled, since the Helmholtz apparatus can control the components of the magnetic field independently and because of the design of the robot. This is in particular discussed in

page 5 of the supporting text, however we do not see experimental evidence of this beside the claims of the authors. Specifically, can the author show evidence that the tail does not move laterally when bent vertically (to trigger the motion) while the robot is rotated horizontally?

The point of this observation is related to a clever design choice, which perhaps the authors do not emphasize enough in the text: the magnetic tail is placed along the middle axis of the robot, and designed to hit the water surface upon magnetic triggering at the center of the rear part of the robot. There the roughly hemispherical end part of the robot confines the spreading Marangoni front of the surfactant and forces directional (forward) propulsion of the robot. However, it seems to us that strictly forward motion is only possible if the tail hits the water exactly at the center (along the middle axis of the robot); if this does not happen - for instance because of deformation or defect of the tail, or indeed because of lateral magnetic deflection - the tail hits the wafer off center, and we could expect the robot not to move forward but rather along a diagonal trajectory. Can the authors substantiate and comment on this aspect?

Related to the previous points, it seems to us that the length of the tail is a prominent parameter. Did the author test the impact of the length of the tail that is submerged in water? This may work as the fin of boats in determining the direction of motion. Is there a maximum submerged length for the tail before the performance of the robot is affected?

The authors show clear evidence of the outstanding performance of the robot in speed and control in confined environment. They also state that the robot control is open loop and programmed. The speed seems so high that, on the other hand, real-time (unprogrammed) manual control of the robot might be impractical; also, at such high speed the robot might easily exit the magnetic domain of the Helmholtz setup if not controlled. So we wonder how the speed can be (down)regulated, in a general case. How can the speed be tuned, and in particular limited? Does one need to act on the surfactant only, or are there other ways to achieve it?

In this line of thought, it seems from the text that the braking is all or nothing. Can the braking also be modulated, so that the robot is still left with a fraction of the speed it had before braking?

Reviewer #2 (Remarks to the Author):

The paper describes a milli-scale robot driven by Marangoni force on the water surface. The robot is controlled by the external magnetic field and performs fast forward and steering movement. The result shows that the robot can move on water with high speed, and navigate with the preprogrammed path. The paper is well-written and includes a wealth of contents and materials. However, it is hard to find academic contribution in terms of new scientific findings, theoretical knowledge for resolving existing problems, except for the technical implementation and demonstration. In addition, the bio-inspiration from *Stenus* gives insufficient motivation for robot design. The main issue is too specific for the broad audience. The paper shows great results, but is dealing with the technical aspects. Reviewer recommends submitting the paper to the more specialized journal in the field. Here are some major concerns on the manuscript before the detail review.

- The title has 'Stenus-inspired', but weak motivation from the organism, except for the abdomen motion. Marangoni propulsion is a well-known phenomenon.
- The paper includes lots of contents and materials. It needs to specify the tackling issue or existing problems on the field for the broader audience.
- It would be better to clarify lessons from this robot or a new enabling technology for this robot.
- Please reduce rhetoric words, and keep objective tone all over the manuscript.

Reviewer #3 (Remarks to the Author):

The authors have developed the Uni-SoPro, a Marangoni effect microskimmer that shows exceptionally high relative accelerations, speed, and decelerations. The Uni-SoPro is inspired by the rove beetle that dispenses a surfactant from its tail when dipped in the water (often to escape predators). Overall, the paper presents a lot of data and I find the fiber dispensing and braking interesting. I think the paper could be significantly improved through a more thorough discussion of motivation and methods, as well as a discussion of how these results could translate to other systems.

Many of the robots/organisms that the authors compare to in Fig 1, Table S1, and the introduction to motivate their work are land-based and therefore not a very good point of comparison given the different physics involved. I'm not familiar with the term 'comprehensive kinematic performance index' (I didn't find this with google so maybe it is defined by the authors?), but if you really want to compare to organisms with higher relative accelerations and velocities, check out LaMSA organisms (e.g., [<https://www.science.org/doi/full/10.1126/science.aao1082>] — there is a table in this paper). In addition, there are several recent papers (including review papers) that cover many of the same kinds of ideas for liquid-based small-scale robots including reconfigurability, systems that utilize the Marangoni effect in a controllable manner, and micro jets.

* [<https://pubs.rsc.org/en/content/articlehtml/2023/sm/d2sm01468h>](DOI: 10.1039/D2SM01468H)

* Review paper: [<https://onlinelibrary.wiley.com/doi/full/10.1002/adma.202205732>]

* Some of the jet-based microswimmers are a better point of comparison and should certainly be discussed. Medina-Sánchez, Mariana ; Magdanz, Veronika ; Guix, Maria ; Fomin, Vladimir M. ; Schmidt, Oliver G. *Advanced functional materials*, 2018, Vol.28 (25), p.1707228

It should be noted that in many of the chemically powered options, an important metric is how long they can be operated for. I don't recall seeing this metric in the authors' work.

How was kinematic performance measured and what is the uncertainty in these measurements? It looks like some of the measurements were taken at 2000 fps but this is not clear in the kinematic characterizations section (different from kinetic which often includes dynamics btw). I'm not familiar with and couldn't find any information on the Phantom R1212 camera. The Phantom V1212 camera is capable of full resolution at much higher frame rates (over 12000 fps) which would provide a much more accurate measurement of acceleration. Similarly, what is the lens used on this camera? Any magnification? It would be useful to know the positional error. Similarly, how many trials were used to determine these measurements?

The subsequent acceleration peaks in Fig 3B bother me — where are these coming from? Are they an artifact of the measurement or is the motion of the Uni-SoPro so variable? If you have a couple trials, can you compare them? If it exists on all of them, then why? Why does it not simply accelerate smoothly and hit a peak speed? Also, in these measurements, does it even hit its peak speed? It seems to still be going up?

In a similar vein, a Helmholtz coil system has constant magnetic field in the center, but a gradient towards the edges. From the movie, I could not tell exactly how far the Uni-SoPro was traveling in each experiment and what this means for the field gradient that is present towards the edges of the coil system. For example, the X and Y positions in Fig 3F seem to extend to the edge of the coils. Was there a specific goal trajectory in this figure?

I find the addition of fibers into the tail intriguing — this seems to be an interesting point. While Fig 2F discusses the tuning capability, I would be interested in seeing how this relates to the incredible accelerations and decelerations seen in the final system. Speaking of which, I'm not sure how tuning capability is defined in the text or in Fig 2F.

I'm not sure how the measurement in Fig 3E was done? Was the UniSoPro turned around each time? Was the amount of surfactant consistent between trials? Is the 'without fiber' decreasing because it releases most of its surfactant at the beginning?

I found the dynamic scenario somewhat challenging to understand. While the text and Fig 7 indicates that the Uni-SoPro is responding to these events, I assume that it is either programmed into the triggering of the Helmholtz coil or a person is triggering the different behaviors. As such, everything is pre-programmed in this scenario. If I got this wrong, then much more discussion of the feedback mechanism is needed for clarity. If this is a correct interpretation, then I think the way it's being told is overselling the situation and can be very confusing for a reader.

The authors state that a magnetic tail with a 'longer length, finer caliber, and lower magnetic particle count exhibited a more favorable deflection response...' — There is really nothing in Figs B-D that is at all surprising and if trying to condense, this could be cut. There is no real optimal found here — just expected trends.

The discussion section could use some further discussion of how these results could be translated to other systems. For example, could the braking ideas be useful for the numerous other means for controlling the Marangoni effect discussed in the papers cited earlier in this review? Is this only useful in a Helmholtz coil system or could it be useful in some more autonomous water skimming robots at some point?

There is significant emphasis on being soft, especially in the abstract and introduction, but it is unclear what the importance of this is besides reconfiguration (which can be done in both more rigid, jointed mechanisms as well as soft mechanisms)

Why is the speed not = 0 after braking in Movie S2?

For scaling, were these geometrically scaled? What dictated the shape of the Uni-SoPro?

I appreciate that it is challenging to write papers in English when it is not one's native language. I highly recommend using some of the AI grammar engines that exist now or a native speaker for proofing however — the paper was challenging to read. There are numerous typos throughout.

Response to Comments on “NCOMMS-23-31144”

We extend our sincere gratitude to the three reviewers for their thorough review of our manuscript and for providing us with invaluable comments that have significantly contributed to the enhancement of our work's quality. These comments have not only allowed us to present technical details more meticulously but have also guided us in elevating and elucidating the overall significance of the article, thus broadening its more general impact.

We have conscientiously studied each constructive comment and suggestion, and these constructive comments and suggestions are fully addressed point-by-point with key responses underlined in this letter. To facilitate the reviewers' understanding, we have indicated the corresponding modifications in our manuscript using blue font. The substantial modifications elevate the quality of our manuscript, making it more suitable for publication in *Nature Communications*.

Once again, on behalf of all the authors, we wish to convey our heartfelt appreciation to the reviewers for their diligent efforts in providing valuable feedback. Please find our comprehensive responses to the reviewers' comments provided below:

Reviewer #1

General Comments. The manuscript describes a novel unthetered bio-inspired robot of millimetric size capable, under external magnetic control, of maneuvers over the surface of water of unprecedented dexterity and performance. While the propulsion method is not novel, as it is based on a classic Marangoni effect, the device is innovative for a set of elements: unprecedented speed and acceleration, which competes and even surpass that of the original biological inspiration; controlled delivery of the propellant by magnetically-controlled trigger as well as bio-inspired surfactant delivery mediated by fibers; superhydrophobic body, which prevents the robot from being submerged and as importantly allows the braking maneuver; the braking maneuver itself, which involves magnetic control of the pose of the robot and transfer of its kinetic energy to the surrounding water. As a result, the robot is capable of navigating structured and unstructured water domains at unprecedented speed and with high degree of control and precision under open-loop programmed control. The manuscript is supported by extensive supplementary material including a substantial supplementary text and very expressive videos (which also provide frames included in some of the illustrations).

The work is substantiated by a commendable amount of work - including analytical and FEM modeling, material characterization, high-speed experimental tracking - that elucidates relevant details in most of the aspects of the research. The work stands out in the field of small-scale robotics as ingenious, innovative, and of high quality overall. We support its publication provided the points raised below are addressed.

Response to General Comments. We extend our sincere gratitude to Reviewer #1's positive and constructive comments. We have diligently and meticulously addressed each of the reviewer's concerns and comments point by point, resulting in a significant enhancement in the quality of our work. To facilitate easy tracking, we have marked the corresponding modifications in our manuscript in "blue".

Comment 1.

The quality of the work is not reflected in the quality of the writing. The main text could be more concise in several places, in particular it makes frequent claims on the importance of bio-inspiration and other general comments, which are legitimate but render the text verbose, and this should be avoided. Importantly, English phrasing and grammar and the use of some words (such as "livings", "power(ing) materials") needs a thorough revision throughout the text. By way of text formatting, most of the symbols in eq. 1 to 3 are not defined in the text (and only some in the supporting text). By the same token, the explicit reference "the previous Science Robotics perspective and viewpoint articles" in line 352 should be removed.

Response to Comment 1. We sincerely appreciate your valuable suggestions. We regret that language limitations, as non-speakers, may have affected the presentation of our work. In response, we've taken significant steps to enhance the quality of our manuscript. We've thoroughly revised the main text, engaging the services of Nature Research Editing to improve language, phrasing, and grammar throughout the entire manuscript, leading to a substantial enhancement in overall writing quality.

In addition, we have also condensed the main text to make it more concise and reader-friendly. To facilitate better comprehension, we've supplemented essential symbol definitions for equations 1 to 3. Furthermore, we've removed explicit references to ensure a more streamlined presentation. These efforts align with your recommendations, and we believe they significantly strengthen the manuscript.

Revisions in the manuscript

[where ρ_f is the density of water, g is the gravity acceleration, $h(x)$ is the depth at x , $l(x)$ is the length at x , and β is the angle between the bottom and horizontal plane, μ_f is the viscosity of water, S is the wetting area of the bottom, dv/dz is the velocity gradient at the bottom surface of the Uni-SoPro, γ is the magnitude of the tension locally, \boldsymbol{t} is the unit vector of the direction of tension, \boldsymbol{v} is the velocity of the Uni-SoPro, R_w is the wave resistance.]

Comment 2.

What software was used for the CFD analysis? This is not reported in page 11.

Response to Comment 2. Thank you for your valuable comments, we have supplemented the analysis software (Ansys Fluent, Inc., USA) and post-processing software (Tecplot, Inc., USA) for data statistical analysis and visualization in the corresponding paragraph.

Revisions in the manuscript

[**Computational fluid dynamics analysis.** To comprehend and illustrate the braking mechanisms of Uni-SoPros, the commercial software Ansys Fluent (ANSYS, Inc., USA) is utilized for computational fluid dynamics (CFD) simulations to quantitatively analyse its hydrodynamics with a specifically constructed model. By utilizing a CFD method, we calculated its hydrodynamic drag and energy transfer during the whole moving process. Specifically, a six-degree-of-freedom (6-DOF) model was employed to compute and analyse the kinetic characterizations of Uni-SoPro. In this model, the magnetically/hydrodynamically induced deformation of Uni-SoPro was ignored, the whole body was considered rigid. However, the elastic effect was still considered in the load conditions. The mass, moment of inertia, magnetic moment, and Marangoni force in each motion state were defined in the user-defined functions (UDF). The commercial post-processing software Tecplot 360 (Tecplot, Inc., USA) was utilized for statistical analysis and visualization of the data. Detailed settings are shown in Supplementary Note 6.]

Comment 3.

In Figure 16f the exponential fitting for the case of the tail with fiber 600 ms does not seem accurate and fair, in fact the STTC for the 5th and 6th set seem comparable and higher than for the fiber cases.

Responses to Comment 3. Thank you for your valuable comments. As you rightly pointed out, the fitting curve cannot well fit all the data, since there are some variations in data especially for the case of the tail with fiber 600 ms and 1000 ms. However, based on the statistical results of surface tension tuning experiments, we can find that there was a dramatically degressive local tension tuning capability, as the trigger number increased when the fiber was not inserted. By contrast, that with fibers shows a slow and linear-like attenuation of local tension tuning, particularly at 200 ms trigger time. In this case, it is difficult to fit all the data well unless overfitting. Therefore, we have opted to remove the fitting curves and present only the original data.

Revisions in the manuscript

Main Figure:

Figure R1. Revised Fig. 2f that removed the fitting curves.]

Comment 4.

In line 122-123 and figure 2D the authors show that a lower magnetic particle concentration in m-PDMS improves the performance of magnetic control. We wonder what is the reason for this, is that because of the lower stiffness of the magnetic tail for lower particle density? Did the author detect a lower limit of particle density in the tail that still makes the robot functional?

Responses to Comment 4. Thank you for your professional comments, we strongly agree with your comments and your analyses regarding this phenomenon. According to our analysis, there are two main potential reasons that may account for such a phenomenon: First, as shown in the following process flow diagram of the magnetic tail (Fig. R2), we fabricated such magnetic thin pipes through (iii) spin coating or natural placing. Therefore, under the same speed of spin coating (RPM) or natural placing condition, due to the viscosity differences caused by different particle concentrations, the thickness of the lower particle density will be thinner, which will significantly influence the moment of inertia ($I_x = \pi(D^4 - d^4)/64$, where D is the outer diameter that depends on the viscosity (positive correlation with the particle concentration) and spin coating speed, d is the inner diameter that depends on the needle tube outer diameter, I_x is the moment of inertia of the cross-section on the x-axis). Such a viscous effect coupled with particle concentration during fabrication would significantly influence the deflection behavior (bending stiffness) (The bending stiffness of the beam can be calculated as $K_b = 3EI_x/L^3$, where E is the young's modulus, L is the length of the magnetic tail (L) [Mechanics of hard-magnetic soft materials. *J. Mech. Phys. Solids.* **124**, 244–263 (2019)]). Second, by carefully checking the tested specimens, we found that the used fixture has shrunk the magnetic tail at the fixed end and formed a crease in the thinner tail (lower particle density) in the previous characterizations, which reduced the bending stiffness of the magnetic tail and has caused systematical error.

Figure R2. Fabrication flow diagram of the magnetic tail.

To prevent the crease and make the bending characterizations constant with the actual situation, we assembled the magnetic tails on the fuel tank and recharacterized the deflection behavior. In this characterization, we found that the tail with a too low concentration of magnetic particles would cause fuel delivery (contacting water surface) and desorption (cut off fuel delivery) process failure, as shown in Figure R3. In this work, to ensure robustness, a higher concentration (PDMS: MPs = 21: 40) was adopted for the experiments.

Figure R3. Characterizations of magnetic tails. Effect of mixture ratio of PDMS and magnetic particles (MPs), magnetic tail size scale on the bending response under various external applied magnetic field B_z . Error bars indicate the standard deviation for $n=3$ sample measurements at each data point. **d** Detaching behavior of the magnetic tail from the water surface with different mixture ratio of PDMS and magnetic particles.

We sincerely appreciate your valuable comments and feedback. Based on the new results obtained, we have incorporated a comprehensive discussion into the relevant sections of the original text. These additional characterization results and discussions further enhance the overall quality and completeness of our manuscript.

Revisions in the manuscript

[Main text: We examined the influence of the magnetic particle content and magnetic tail size scale on its behaviours of downward bending for fuel delivery and upward bending for detaching from the water surfaces, as shown in Fig. 2b-d. Our findings revealed that a magnetic tail with higher magnetic particle contents exhibited more favourable bending behaviours for ensuring effective fuel delivery to the water surfaces and on-demand cutting of the fuel supply by overcoming the solid-liquid interface energy. In addition, we also

characterized the influence of different postures and submerged magnetic tail lengths on swerving behaviour (Supplementary Fig. 7). The results show that excessive immersion of the magnetic tail in water will attenuate its swerving behaviour. Thus, selecting a suitable magnetized tail with appropriate magnetic particle contents, tail size scale and submerged length of tail is very significant for achieving efficient manoeuvrability.

Main Figure:

Fig. 2. Systematical bio-inspiration from its natural counterparts (*Stenus comma*), and codesign of the soft propulsor with detailed characterizations. **a** Systematical bio-inspiration designs from macro to micro as well as artificial codesign to enable the presented soft propulsor with triggerable tail, stable fuel transport and delivery, non-wetting surfaces, and decoupled steerable capability. **b-c** Effect of mixture ratio of PDMS and magnetic particles (MPs), magnetic tail size scale on the bending response under various external applied magnetic field B_z . Error bars indicate the standard deviation for $n=3$ sample measurements at each data point. **d** Detaching behavior of the magnetic tail]

Comment 5.

The authors claim that the horizontal (swerve, pose) and vertical (trigger, propulsion) control of the robot can be and in fact are decoupled, since the Helmholtz apparatus can control the components of the magnetic field independently and because of the design of the robot. This is in particular discussed in page 5 of the supporting text, however we do not see experimental evidence of this beside the claims of the authors. Specifically, can the author show evidence that the tail does not move laterally when bent vertically (to trigger the motion) while the robot is rotated horizontally?

Responses to Comment 5. Thank you for your professional comments. Theoretically, the manipulation of swerving and triggering motion is decoupled. Following your constructive advice, we supplemented necessary experiments to verify this independent manipulation characteristics of vertical motion of magnetic tail and horizontal rotation of the Uni-SoPro.

Specifically, we manipulated a Uni-SoPro with characteristic length of 7.2 mm to rotate while the magnetic tail is deflected vertically downwards and upwards and recorded whole processes with a high-speed camera at 2000 fps, as shown in Figure R4. It could be found that whether the magnetic tail is bent downwards or upwards, it essentially does not result in a significance deviation (\$< 4^\circ\$ ) from the main axis of the body when the Uni-SoPro is rotating.

In addition, the reviewer must be an expert in this field. In fact, lateral deflection can occur in some certain extreme conditions, for instance, if the bending stiffness was small enough. Within the context of our present design and configuration, it can be effortlessly decoupled to facilitate manipulation.

Figure R4. Decoupled magnetic tail vertical motion and Uni-SoPro's rotation

To make this mechanism clear to readers, we combined this experiment in the previous Supplementary Figure 5, and added corresponding discussions in the manuscript.

Revisions in the manuscript

[Main text: Moreover, to decouple the propulsion and steering control, a naturally decoupled design and manipulation mode were adopted (as shown in Supplementary Fig. 5 and Supplementary Note 1). During the decoupling verification experiments, it was observed that whether the magnetic tail is bent downwards or upwards. The tail essentially does not lead to a significant deviation ($< 4^\circ$) from the main axis of the body when Uni-SoPro is in rotation (Supplementary Movie 4). Ultimately, such a systematic bio-inspiration and comprehensive codesign of the above aspects provide us with a potential embodiment to achieve biological-level and even over-biological motion behaviours.

Supplementary Figure:

Supplementary Figure 4. Decoupled control. (a) Typical decoupled mechanism and typical actuation states. The magnetic tail deflection control is decoupled from Uni-SoPro's swerving as due to structural and magnetic domain presets. The horizontal component B_{x-y} ($B\sin\phi$) determines the direction of the Uni-SoPro, while the vertical component B_z ($B\cos\phi$) determines the bending/tilting degree of the tail. By controlling these two components, B_{x-y} and B_z , respectively, Uni-SoPros can obtain determinant direction and propulsion state concurrently. (b) Schematic diagram illustrating the horizontal deflection of the tail which is concurrently deflected vertically either downwards or upwards, as the Uni-SoPro undergoes horizontal rotation. (c-d) Mode 1: The tail horizontal deflection when bent vertically downwards, and (e-f) Mode 2: The tail horizontal deflection when bent vertically upwards, both while the Uni-SoPro is rotated horizontally (from 0° to 90°). (g-h) Keyframes of decoupling motion of tail vertical deflection and Uni-SoPro's rotation. The characteristic length of Uni-SoPro is 7.2 mm in this experiment.]

Comment 6.

The point of this observation is related to a clever design choice, which perhaps the authors do not emphasize enough in the text: the magnetic tail is placed along the middle axis of the robot, and designed to hit the water surface upon magnetic triggering at the center of the rear part of the robot. There the roughly hemispherical end part of the robot confines the spreading Marangoni front of the surfactant and forces directional (forward) propulsion of the robot. However, it seems to us that strictly forward motion is only possible if the tail hits the water exactly at the center (along the middle axis of the robot); if this does not happen - for instance because of deformation or defect of the tail, or indeed because of lateral magnetic deflection - the tail hits the wafer off center, and we could expect the robot not to move forward but rather along a diagonal trajectory. Can the authors substantiate and comment on this aspect?

Responses to Comment 6. Thank you for your valuable and professional comments. We quite agree with your positive recognition on our design. Indeed, the motion of Uni-SoPro is highly related with the fuel delivery strategy, much akin to the principle of a vector engine. In our current design, the magnetic tail is placed along the middle axis of the robot. Therefore, the Uni-SoPro will only move forward, while the direction is controlled by the active steering (This case is adopted by this work, small assembly errors will not have a significant impact on the direction/trajectory under the magnetic steering). But if the fuel delivery point were to be positioned off-center on the Uni-SoPro, this would cause an asymmetric tension force distribution along the contact line and thus lead to a diagonal motion in the same magnetic manipulation condition.

To substantiate this point further, we assemble an asymmetric Uni-SoPro with an obliquely mounted magnetic tail (the angle between the magnetic tail and the main axis is $\sim 10^\circ$) to test such a case, Fig. R5 and Supplementary Movie 4. This experiment further affirmed that such a configuration does, indeed, produce a

diagonal trajectory under normal magnetic triggering conditions (direction control magnetic field component B_{x-y} is aligned with the B_x), aligning exactly with your analysis.

Figure R5. Propulsion test with an off-center magnetic tail

Following your advice, we emphasize the design strategy in the manuscript and supplemented the corresponding experiment details in the Supplementary Materials.

Revisions in the manuscript

[Main text: First, at the macroscopic scale, we noticed that the flexible bendable abdomen in *Stenus comma* is a critical structure that delivers propulsion materials on the water surfaces on demand. By virtue of the fast response of magnetic manipulations^{18,19}, a magnetic tail was introduced along the middle axis to mimic *Stenus*' flexible swung abdomen (controlled via a programmable three-dimensional Helmholtz coil manipulation platform, as shown in Supplementary Fig. 1). Such a design ensures movement along the primary axis and enables on-demand, highly dynamic manipulation of its propulsion. In cases where fuel delivery is off-centre, it would result in an asymmetric tension force distribution along the contact line, thereby inducing diagonal motion facilitated by magnetic navigation (as shown in Supplementary Fig. 2).

Supplementary Figure:

Supplementary Figure 2. Propulsion test with an off-center magnetic tail. The Uni-SoPro with an obliquely mounted magnetic tail (the angle between the magnetic tail and the main axis is $\sim 10^\circ$) produces a diagonal trajectory under a normal manipulation magnetic fields (direction controlling magnetic field B_{x-y} is aligned with the x axis). The characteristic length of the Uni-SoPro in this test is 7.2 mm.]

Comment 7.

Related to the previous points, it seems to us that the length of the tail is a preminent parameter. Did the author test the impact of the length of the tail that is submerged in water? This may work as the fin of boats in determining the direction of motion. Is there a maximum submerged length for the tail before the performance of the robot is affected?

Responses to Comment 7. We sincerely appreciate your valuable and professional comments, and we wholeheartedly agree with your assessment of the significance of tail length as a critical parameter in our study. Following your guidance, we have undertaken additional comparative experiments to investigate the impact of tail submersion length, which is controlled by the magnetic field angle φ . Specifically, when φ falls within the range of 90 degrees to 180 degrees, the submerged length of the tail increases as φ increases. We have observed that excessive immersion of the magnetic tail in water has a notable effect on its swerving behavior, Figure. R6. In particular, as φ approaches 170 degrees, we have observed a significant attenuation in the swerving behavior.

Figure R6. The influence of different postures and submerged magnetic tail length on its swerving behavior. A Uni-SoPro with a size scale of 1.4 rotates under the magnetic fields with different φ . The φ mainly affects the deflection of the magnetic tail and the length of immersion in the water.

Thank you for your valuable comments and feedback. Based on the new results obtained, we have incorporated a comprehensive discussion into the relevant sections of the original text. These additional

characterization results and discussions further enhance the overall quality and completeness of our manuscript.

Revisions in the manuscript

[Main text:

In addition, we also characterized the influence of different posture and submerged magnetic tail length on its swerving behavior, Supplementary Fig. 7. The results show that excessive immersion of the magnetic tail in water will attenuate its swerving behavior. Consequently, selecting a magnetized tail with appropriate magnetic particle contents, tail size scale and submerged length of tail is of significance in achieving efficient maneuverability.

Supplementary Figure:

Supplementary Figure 7. The influence of different postures and submerged magnetic tail lengths on its swerving behaviour. A Uni-SoPro with a size scale of 1.4 rotates under the magnetic fields with different φ . The φ mainly affects the deflection of the magnetic tail and the length of immersion in the water.]

Comment 8.

The authors show clear evidence of the outstanding performance of the robot in speed and control in confined environment. They also state that the robot control is open loop and programmed. The speed seems so high that, on the other hand, real-time (unprogrammed) manual control of the robot might be impractical; also, at such high speed the robot might easily exit the magnetic domain of the Helmholtz setup if not controlled. So we wonder how the speed can be (down)regulated, in a general case. How can the speed be tuned, and in particular limited? Does one need to act on the surfactant only, or are there other ways to achieve it?

Responses to Comment 8. Thank you for your professional comments. We quite agree with you that real-time speed tuning is a very meaningful aspect of such propulsion mode. However, due to the limited resolution (~100 ms) of the current manipulation platform and the absence of an effective mechanism for precise speed regulation, achieving accurate real-time tuning remains a challenge. This aspect could be addressed in future work, since the key contribution of this work is to address *Stenus*-inspired on-demand and high-performance propulsion and braking. For the most cases, we usually execute a braking command in the end to avoid its collisions with boundaries.

In addition, for the real-time speed regulation, we also considered many strategies and conducted a series of experiments based on our current conditions. Here, we provided two available strategies for equivalent speed control: intermittent propulsion and zigzag propulsion, as shown in Figure R7.

Figure R7. Two equivalent speed regulation strategies. (a) Schematic depicting the concept concepts of intermittent propulsion (mode 1) and zigzag propulsion (mode 2). In mode 1, speed control is achieved by manipulating the speed attenuation coefficient R_i , which is proportionally related to $T_p / (T_p + T_b)$. In mode 2, speed regulation relies on the speed attenuation coefficient R_z , which exhibits a positive correlation with $\cos\theta_z$. (b) The speed regulation for a Uni-SoPro with a scale of 1.2 in intermittent propulsion ($T_p / (T_p + T_b) = 0.5$) and in zigzag propulsion ($\cos\theta_z = 0.5$).

These two speed regulation strategies can be achieved by fine-tuning R_i and R_z through precise control of magnetic fields. As confirmed through experimental validation, the speed can be effectively decreased or regulated by manipulating magnetic fields. Additional results supporting these findings have been included in both the main text and supplementary materials.

Revisions in the manuscript

[Main text: In addition, we developed two equivalent speed regulation strategies, namely, intermittent propulsion and zigzag propulsion, as illustrated in Supplementary Fig. 19a. As confirmed via experimental validation, by tuning R_i and R_z through controlling magnetic fields, the speed can be equivalently regulated, as shown in Supplementary Fig. 19b.

Supplementary Figure:

Supplementary Figure 18. Two equivalent speed regulation strategies. (a) Schematic depicting the concept concepts of intermittent propulsion (mode 1) and zigzag propulsion (mode 2). In mode 1, speed control is achieved by manipulating the speed attenuation coefficient R_i , which is proportionally related to $T_p / (T_p + T_b)$. In mode 2, speed regulation relies on the speed attenuation coefficient R_z , which exhibits a positive correlation with $\cos \theta_z$. (b) The speed regulation for a Uni-SoPro with a scale of 1.2 in intermittent propulsion ($T_p / (T_p + T_b) = 0.5$) and in zigzag propulsion ($\cos \theta_z = 0.5$.)]

Comment 9.

In this line of thought, it seems from the text that the braking is all or nothing. Can the braking also be modulated, so that the robot is still left with a fraction of the speed it had before braking?

Response to Comment 9. We sincerely appreciate your valuable comments, and we fully concur with your insight that precise braking modulation is as meaningful as the previous comment regarding real-time speed tuning. Our extensive experiments have revealed a correlation between braking behavior and posture changes, which can be modulated by the magnetic field. Illustrated in Fig. 4b-d, a series of gradient braking experiments with different magnetic field parameters was conducted. The results indicate that both residual velocity and velocity rebound are influenced by the dynamics of posture changes controlled through real-time magnetic fields. However, the limitations of our current manipulation platform, which cannot generate a finer magnetic field to induce more complex dynamics of posture changes, pose a challenge in achieving precise braking modulation over a large range.

We believe that achieving better and more precise braking modulation over a large range is feasible with a highly responsive magnetic manipulation platform. Theoretical considerations suggest a strong correlation between waving resistance and the dynamics of posture changes of Uni-SoPro. In our current conditions, however, we have observed that this braking behavior can only be tuned through the manipulation of magnetic fields within a limited range. The full potential of precise braking modulation may be realized with an advanced magnetic manipulation platform.

Fig. 4. Braking hydrodynamics analysis. *b* Experimental verification of the whole braking process under different braking magnetic fields. *c* Full cycle record of the position and attitude of Uni-SoPro in the above experiments. The time interval of sampling points is 1.25 ms in the braking process and 6.25 ms in the cruising process. *d* Full cycle record of the velocity and attitude of the Uni-SoPros, and the results show a high relevance between the velocity and attitude. *e* The changing of subjected forces in different states (from state S_1 to state S_4).

To provide a lucid elucidation of these findings, we have thoughtfully incorporated essential clarifications within the manuscript.

Revisions in the manuscript

[Main text: To gain deeper insights into the braking dynamics, we conducted a series of gradient braking experiments. These experiments involved the application of magnetic fields with identical magnitudes (100 Gs), albeit with varying angles (defined as the included angles of Magnetic Field \mathbf{B} and z -axis, denoted as φ , and here, 10° , 45° , and 60° were selected), as shown in Fig. 4b. As observed, different included angles of the magnetic fields lead to different posture changes and trajectories (Fig. 4c). By extracting key parameters (the velocity and titling angle β) of Uni-SoPros during the process, as shown in Fig. 4d, we observed that the velocity attenuation and subsequent rebounding are highly related to the external magnetic field. Such a braking behaviour can be tuned through the manipulation of magnetic fields within a small range on our current manipulation platform.]

Once again, we sincerely appreciate your highly positive recognition and support on our work. Your professional and invaluable comments have provided us with insightful guidance, further greatly enhancing the overall quality of this work. These significant improvements render this work more suitable for publication in this journal.

Reviewer #2

General Comments. The paper describes a milli-scale robot driven by Marangoni force on the water surface. The robot is controlled by the external magnetic field and performs fast forward and steering movement. The result shows that the robot can move on water with high speed, and navigate with the preprogramed path. The paper is well-written and includes a wealth of contents and materials. However, it is hard to find academic contribution in terms of new scientific findings, theoretical knowledge for resolving existing problems, except for the technical implementation and demonstration. In addition, the bio-inspiration from *Stenus* gives insufficient motivation for robot design. The main issue is too specific for the broad audience. The paper shows great results, but is dealing with the technical aspects. Reviewer recommends submitting the paper to the more specialized journal in the field. Here are some major concerns on the manuscript before the detail review.

Response to General Comments. We extend our heartfelt gratitude to the reviewer for your comprehensive evaluation and for raising insightful concerns and comments from a high-level perspective. We wholeheartedly agree with the significance of these concerns, which is exactly what we're pursuing and have devoted considerable effort to over the past a few years. As underscored by the reviewer, the preeminent contributions and significance of this work lie the emergence of new scientific findings, the advancement of theoretical knowledge, and the motivation for robot design.

The authors would like to appreciate the reviewer for raising the concern, which we believe is mainly caused by misunderstandings that suggest we need to improve our presentation. There are indeed quite a few new scientific findings, theoretical studies, breakthrough in performance results as well as the broader implications of generalizable bionics on the design of robots, not just a technical implementation/solution. Please thus allow us to clarify point by point.

1. Scientific findings and theoretical knowledge

A new design achieves a magnitude of order performance/controllability improvement than previous approaches, indicating that there must be scientific principles behind. We also devoted an effort on investigating the scientific principle behind it. Here are three points that highly relevant to scientific discoveries and theoretical knowledge:

(i) (a) A tension gradient induced and (b) Marangoni flow-induced momentum transfer hybrid actuation principle (previous similar works are based on Marangoni effect) was found, enabling the Uni-SoPro to reach a CKPI of $1.6 \times 10^6 \text{ BL}^2/\text{s}^3$, and according to the literature survey, it surpasses current state-of-the-art (with a CKPI $\sim 6.1 \times 10^4 \text{ BL}^2/\text{s}^3$) by several orders at the same scale and is comparable to fast-moving insects in similar moving modes, e.g., *Stenus*, and even superior to that of some other fast-moving natural livings moving on water, e.g., water skis (with a CKPI, $\sim 8.4 \times 10^4 \text{ BL}^2/\text{s}^3$);

Diverging from previous Marangoni-based actuation methods, we found relying solely on tension cannot generate such a large acceleration performance. Therefore, we conducted an analysis of the propelling principles in the whole process. Notably, based on the experimental data and theoretical calculations, we conducted an in-depth analysis of the different forces acting on Uni-SoPros during each state (Fig. R8).

Figure R8. The changing of subjected forces in different states. Marangoni flow-induced momentum transfer plays a synergistical role in initial propelling process (S1: Cruising state)

Our investigation revealed the presence of several dominant forces such as momentum, hydrostatic pressure, viscous force, wave resistance, and tension that govern the overall process, which can be effectively described by the following hydrodynamic equation,

$$F = \int \underbrace{\rho_f g h(x) l(x) \sin \beta dx}_{\text{hydrostatic}} + \underbrace{\mu_f S \frac{dv}{dy} \cos \beta}_{\text{viscosity}} - \underbrace{\int_C \gamma t \frac{v}{|v|} dl}_{(a) \text{ tension}} + \underbrace{R_W}_{\text{wave resistance}} - \underbrace{\rho_f S_1 (v_c - v)^2 \sin \beta}_{(b) \text{ momentum}}. \quad (1)$$

Our experiment analysis and calculations unveiled a new mechanism, showcases the simultaneous collaboration of two primary driving forces (tension gradient induced and Marangoni flow-induced

momentum transfer) during the Uni-SoPro's initial acceleration phase. This hybrid actuation principle elucidates the underlying reasons behind the exceptional acceleration performance exhibited by the Uni-SoPro.

(ii) Another interesting scientific finding is that we found with such a bioinspired design with the intervene of the artificial structure, the Uni-SoPro can achieve a beyond biological-level braking behavior. To analyzing the mechanism and sciences behind this behavior, through systematical experiments, modeling, and computation fluidic dynamics (CFD) analysis, we found that there is a drastic momentum transfer of the kinetic energy to the water environment by rapid posture changing, enables Uni-SoPros with previously inaccessible agility, particularly on braking ($\sim 5,010 \text{ BL/s}^2$).

Figure R9. Computational fluid dynamic (CFD) analysis of the whole braking process

In the CFD analysis, a full-circle kinetic/gravitational energy of Uni-SoPro, surface/kinetic/gravitational energy of water, horizontal drag force, horizontal velocity, and posture of Uni-SoPro were calculated throughout the entire process. The results indicate that the horizontal velocity attenuates within approximately 15 ms, resulting from the rapid change in tilting angle during the braking state. From the back-tilting state to the horizontal state, $\sim 92.5\%$ kinetic energy drastically attenuates and transfers to the kinetic energy of water and the gravitational energy of Uni-SoPro, which agrees well with the finding from experiments, as shown in Fig. R9.

(iii) To make this work more systematic and offer a stronger theoretical framework for manipulation of such propulsors, we formulated a kinematic model, incorporating essential components such as a direction parameter, a typical velocity curve, a distance function and an attenuation function, as outlined in Fig. R10, Eq (2) and (3).

Figure R10. Theoretical kinematic modeling for multi-segment motion trajectory planning.

$$\begin{cases} \theta \\ v_{f1}(t) \\ v_{f2}(t) \\ v_{f3}(t) \\ v_{f4}(t) \end{cases} = \begin{cases} \theta_n & t_0 \leq t < t_4 \\ a_1 - e^{b_1(t-t_0)+\ln(a_1)} & t_0 \leq t < t_1 \\ v_{f1}(t_1) \cdot e^{b_2(t-t_1)} & t_1 \leq t < t_2 \\ v_{f2}(t_2) \cdot e^{b_3(t-t_2)} & t_2 \leq t < t_3 \\ v_{f3}(t_3) \cdot e^{b_4(t-t_3)} & t_3 \leq t < t_4 \end{cases}, \quad (2)$$

$$\begin{cases} \theta = \theta_n \\ v_f(n, t, T_t) = k(n) \cdot v_f(t, T_t) \\ l_n = \int v_f(n, t, T_t) dt \\ k(n) = \begin{cases} e^{-\lambda(n-1)}, (\text{w/o fiber}) \\ \kappa(n-1) + 1, (\text{w/ fiber}) \end{cases} \end{cases}. \quad (3)$$

This model was developed subsequent to a deep comprehension of the propulsion and braking hydrodynamics of the Uni-SoPro, enabling us to program its movement.

2. Motivation for robot design

Actually, our work is trying to solve the grand challenges in the robotic community. As we mentioned in the Introduction section, due to their compact size, making it hard to codesign the whole structure, and thus making untethered insect scale robots difficult to simultaneously achieve high kinetic performance and agility during movement/interaction.

While in Nature, *Stenus commas* (Coleoptera, Staphylinidae/rove beetle) can achieve an instantaneous high speed (400~750 mm/s or 80~150 BL/s) to dash over water surfaces (4) [*Chemical ecology—a chapter of modern natural products chemistry. Angew. Chemie Int. Ed. English. 15, 214–222 (1976)*]. This on-demand, quick-response and high-performance propulsion ability give us great motivation to study the mechanisms behind it. With an in-depth investigation of the structures from microscale to macroscale, we found there are many elegant features that collectively contribute to its exceptional behavior, Fig. R11.

Figure R11. Systematical bio-inspiration designs from macro to micro as well as artificial codesign to enable the presented soft propulsor with triggerable tail, stable fuel transport and delivery, non-wetting surfaces, and decoupled steerable capability.

*i. At macroscopic scale, the flexible bendable abdomen in *Stenus comma* is a critical structure for its on-demand delivering powering materials on the water surface.*

ii. At microscopic scale, countless tiny conducting canals in the gland system serve as fuel transport bridges.

*iii. At mesoscopic scale, with bristles-filled surfaces, *Stenus comma* can prevent from sinking in water, which is of significance for guaranteeing its stability propelling/operating on the water surface.*

The presence of these multiscale structures has proven to be a rich source of inspiration for the design of propulsors at this scale. It emphasizes the significance of simultaneously considering multiple levels or aspects within the constraints of a compact size. This approach enables the seamless collaboration of structures and functions, ultimately leading to the attainment of specific characteristics.

We firmly believe that embracing a system-level perspective to investigate the structures and behaviors of organisms, comprehending the underlying mechanisms, and then collaboratively designing and implementing solutions using existing technologies presents a potent approach. Within this process, the

integration of certain artificial structures and mechanisms can lead to behaviors that surpass those found in their natural counterparts. For example, the incorporation of decoupled embedded steering chips enhances its flexibility in steering, while rapid posture changes facilitate agile braking. This systematic approach to bio-inspiration, coupled with the strategic use of artificial structures and mechanisms, offers a promising paradigm for designing small-scale robots with exceptional performance, even surpassing their natural counterparts.

Biomimetics have indeed brought us many motivations for robot design. we also summarized some similar bioinspired work with excellent performances/capabilities in the following table.

Robot	Main figures/concepts	Main advancements/highlights	Journal
Un-SoPro		Systematically inspired by the swift abdomen swung, built-in secretion gland, secretion transport/deliver, and body setae of Stenus comma , together with magnetic-induced fast-transformed postures, we present a swift, agile untethered millimeter-scale soft propulsor operating on water. The demonstrated one, with a body length (BL) of 3.6 mm, achieved a recorded specific speed of ~ 201 BL/s, acceleration of $\sim 8,372$ BL/s ² , whose comprehensive kinetic performance surpasses that of previous ones in similar scales by several orders. Furthermore, we find a momentum-transfer induced over-biological on-demand braking (with a deceleration $\sim -5,010$ BL/s ²) and reveal its hydrodynamics behind such instant braking.	This work
Robotic insects (5)		They elucidated the hydrodynamics involved and develop a bio-inspired impulsive mechanism maximizes momentum transfer to water.	Science (2015)
Engineered jumpers (6)		They created a device that can jump over 30 meters high, and their work advances the understanding of jumping, shows a new level of performance, and underscores the importance of considering the differences between engineered and biological systems.	Nature (2022)
Three-dimensional electronic microfliers (7)		They demonstrated a range of 3D macro-, meso- and microscale fliers produced in this manner, including those that incorporate active electronic and colorimetric payloads. Analytical, computational and experimental studies of the aerodynamics of high-performance structures of this type establish a set of fundamental considerations in bio-inspired design, with a focus on 3D fliers that exhibit controlled rotational kinematics and low terminal velocities.	Nature (2021)
Self-power soft robot (8)		Inspired by the structure of a deep-sea snailfish, they developed an untethered soft robot for deep-sea exploration, with onboard power, control and actuation protected from extraordinarily high pressure by integrating electronics in a silicone matrix.	Nature (2021)

In terms of motivation, our work introduces a multiscale bioinspired paradigm that facilitates the attainment of locomotion behaviors at both biological and even beyond-biological levels. This achievement is underpinned by a thorough comprehension of the underlying hydrodynamics. We believe that our contribution holds considerable significance for the advancement of insect-soft robots, particularly when contrasted with prior similar endeavors. Moreover, such a systematical bio inspiration design strategy collaborating with other artificial mechanisms/structures is potential extend to the broader realm of generalizable bionics, influencing the design of robots across various contexts.

It's possible that our previous presentation didn't effectively convey these critical points, which could have led to the article not adequately addressing the concerns highlighted by the reviewers. In response, we have embarked on comprehensive revisions throughout the entire text to ensure that the necessary improvements are incorporated.

Comment 1.

The title has ‘Stenus-inspired’, but weak motivation from the organism, except for the abdomen motion. Marangoni propulsion is a well-known phenomenon.

Response to Comment 1. Thank you for your valuable comments. We quite agree with you that Marangoni propulsion is a well-known phenomenon and that is widely utilized for the propulsion of small-scale robots (9) (10) [Bio-inspired untethered fully soft robots in liquid actuated by induced energy gradients, Natl Sci Rev, 2019, Vol. 6, No. 5; Multifunctional and biodegradable self-propelled protein motors. Nat Commun, 10, 3188 (2019)]. However, the novelty in our work lies in achieving on-demand, high-performance propulsion and braking, addressing the existing limitations related to the lack of an on-demand fuel mechanism and instant braking ability.

As you correctly highlighted, the motion of the abdomen is indeed a crucial biomimetic aspect inspired by Stenus. We believe that several other aspects have also significantly inspired us in enabling Uni-SoPro, including the multi-scale structure, propulsion principle, and biological behaviors of Stenus. In this context, we think the term "Stenus-inspired" will be more comprehensive. We will explain each aspect point by point in the following paragraphs:

Multi-scale structures

To enable the stable on-demand propulsion and achieve high performance in Uni-SoPro, our investigation revealed many other elegant features. These features collectively contribute to its exceptional behavior with an in-depth investigation of the structures from microscale to macroscale Fig. R12.

Figure R12. Systematical bio-inspiration designs from macro to micro as well as artificial codesign to enable the presented soft propulsor with triggerable tail, stable fuel transport and delivery, non-wetting surfaces, and decoupled steerable capability.

Besides triggerable abdomen at macro scale for on-demand fuel delivery, for instance, at mesoscopic scale, with bristles-filled surfaces, *Stenus comma* can prevent from sinking in water, which is of significance for guaranteeing its stability propelling/operating on the water surface. We found that the stability was

relatively poor (the overall density of the Uni-SoPro is twice to three times that of water) without optimized design of superhydrophobic microstructure surfaces. In addition, the superhydrophobic surface is also beneficial for reducing the water resistance caused by viscous forces. *At microscopic scale*, countless tiny conducting canals in the gland system serve as fuel transport bridges. Inspired by this, a receiving/conducting canal-inspired surfactant sub-division transport mechanism was introduced to optimize delivery by inserting fiber at the tip of the tail. We have found that with such inserted fiber, the fuel delivery process will be more linear. It offers a relatively ideal trigger attribute by providing a sufficient time window with expectable velocity for precise manipulation, and further trajectory control/planning, as shown in Fig. R13, e-f.

Figure R13. Schematic illustration of the conducting canal-inspired inserted fiber smoothing fuel delivery compared with the one without inserted fiber, and the comparison experiments of multiple local surface tension tuning show that the tail with (w/) inserted fiber shows a linear-like local surface tension tuning capability as the increment of accumulated trigger number, particularly in a short touching mode, while the local surface tension tuning capability (abbreviated as STTC) of that without (w/o) inserted fiber dramatically decreases as the increment of accumulated trigger number, $n=3$ sample measurements for a referenced group (w/o inserted fiber). Error bars indicate the standard deviation for $n=3$ sample measurements at each data point.

Propulsion mechanism

The main propulsion mechanism is similar to that of *Stenus*, involving the delivery of surfactant to influence the surface tension gradient. Our design achieves a magnitude of order performance/controllability improvement than previous approaches, indicating that there must be scientific principles behind. We also devoted an effort on investigating the scientific propulsion principle behind it. Here, we first revealed that a tension gradient-induced and Marangoni flow-induced momentum transfer hybrid actuation principle to enable such a propulsion process.

A tension gradient induced and Marangoni flow-induced momentum transfer **hybrid actuation principle** (previous similar works are based on Marangoni effect) was found, enabling the Uni-SoPro to reach a CKPI of $1.6 \times 10^6 \text{ BL}^2/\text{s}^3$, and according to the literature survey, it surpasses current state-of-the-art (with a CKPI $\sim 6.1 \times 10^4 \text{ BL}^2/\text{s}^3$) by several orders at the same scale and is comparable to fast-moving insects in similar

moving modes, e.g., *Stenus*, and even superior to that of some other fast-moving natural livings moving on water, e.g., water skis (with a CKPI, $\sim 8.4 \times 10^4 \text{ BL}^2/\text{s}^3$);

Diverging from previous Marangoni-based actuation methods, we found relying solely on tension cannot generate such a large acceleration performance according to calculations. Therefore, we conducted an analysis of the propelling principles in the whole process. Notably, based on the experimental data and theoretical calculations, we conducted an in-depth analysis of the different forces acting on Uni-SoPros during each state, as shown in Fig. R14.

Figure R14. The changing of subjected forces in different states. Marangoni flow-induced momentum transfer plays a synergistical role in initial propelling process (S1: Cruising state)

Our investigation revealed the presence of several dominant forces such as momentum, hydrostatic pressure, viscous force, wave resistance, and tension that govern the overall process, which can be effectively described by the following hydrodynamic equation,

$$F = \int \underbrace{\rho_f g h(x) l(x) \sin \beta dx}_{\text{hydrostatic}} + \underbrace{\mu_f S \frac{dv}{dy} \cos \beta}_{\text{viscosity}} - \underbrace{\int_C \gamma t \frac{v}{|v|} dl}_{(a) \text{ tension}} + \underbrace{R_{W}}_{\text{wave resistance}} - \underbrace{\rho_f S_1 (v_c - v)^2 \sin \beta}_{(b) \text{ momentum}}. \quad (1)$$

Our experiment analysis and calculations unveiled a new mechanism, showcases the simultaneous collaboration of two primary driving forces (tension gradient induced and Marangoni flow-induced momentum transfer) during the Uni-SoPro's initial acceleration phase. This hybrid actuation principle elucidates the underlying reasons behind the exceptional acceleration performance exhibited by the Uni-SoPro.

Biological behaviors

From a broader biological perspective, through the systematic biomimicking enabled swift movement and agile action, the presented Uni-SoPro can mimic a predation-escaping behaviour similar to its natural counterpart, as illustrated in Figure R15.

Figure R15. By mimicking a rove beetle's behaviour, a Uni-SoPro is dodging a fast and violent predator (lizard) in a pond

Considering the three aspects mentioned above, we believe that “Stenus-inspired” is a more suitable term for describing the presented Uni-SoPro. Following your valuable suggestions, to clearly express the biomimetic motivation from the *Stenus* and elucidate the novelty of the propulsion principles, we carefully revised the manuscript.

Revisions in the manuscript

[Main text: [Introduction section] Due to its elegant multiscale structural features, including a flexible swingable abdomen, a surface covered in bristles, and specialized conducting canals for fuel transport, *Stenus commas* (Coleoptera, Staphylinidae, commonly known as rove beetles) can achieve an instantaneous high speed (400~750 mm/s or 80~150 BL/s) to dash over water surfaces. Such an agile reaction and swift movement hence enable *Stenus* to quickly escape from dangers or avoid being predated¹⁷.

By systematically mimicking such unique multiscale biological structures, mechanisms, and motion behaviour and combining them with precise yet fast-responsive magnetic manipulation, an engineering approach and corresponding fundamental sciences are presented in this work to realize swift, agile untethered insect-scale soft propulsors (Uni-SoPros), as shown in Fig. 1b, with comprehensive consideration of the following aspects (Fig. 1c). First, a bioinspired real-time magnetic-triggered tail is introduced to on-demand, real-time transfer propulsion materials (surfactants) from a tank to water behind its caudal fin, providing primary controllability. Second, a precise release of surfactants results in a better-controlled surface tension gradient. Such a gradient difference along the contact line directly induces an expected horizontal component of the resultant, pulling Uni-SoPros forward. Simultaneously, drastic Marangoni flow impacts on the caudal fin and induces a momentum transfer to Uni-SoPros, particularly during the initial acceleration state, resulting in a synergistic actuation that guarantees swift motions. Third, by utilizing fast posture transformations, a rapid momentum transfer is induced to water and attenuates the most kinetic energy, enabling Uni-SoPros agile braking in dynamic situations. Consequently, Uni-SoPros exhibits *Stenus commas*-like on-demand controllability and

comparable comprehensive kinetic performance to their natural counterparts, *Stenus commas* (Supplementary Movie 1). Its comprehensive kinetic performance index (CKPI, an all-in-one benchmark defined as a product of the relative peak acceleration, BL/s², and relative peak velocity, BL/s, where the relative performance refers to the ratio of performance to body length) of 1.6×10^6 BL²/s³ surpasses that previously reported by several orders, as shown in Fig. 1d and Supplementary Table 1. Moreover, the presented Uni-SoPro shows an agiler braking capability with a deceleration of -5,010 BL/s² (several orders higher than its natural counterparts), as shown in Fig. 1e and Supplementary Movie 2.

[Braking hydrodynamics section] Particularly, our experimental analysis and calculations unveiled a new mechanism, showcasing the simultaneous collaboration of two primary driving forces (the tension gradient-induced and Marangoni flow-induced momentum transfers) during Uni-SoPro's initial acceleration phase. Such a hybrid actuation principle elucidates the underlying reasons behind the acceleration performance exhibited by Uni-SoPro. In addition, our findings indicate that the dominant force during braking (S3), as illustrated in Fig. 4e and elaborated in Supplementary Note 2, is the momentum transfer-induced force (wave resistance). This critical hydrodynamic mechanism underlies a rapid posture transformation-induced braking of Uni-SoPros.]

Comment 2.

The paper includes lots of contents and materials. It needs to specify the tackling issue or existing problems on the field for the broader audience.

Response to Comment 2. Thank you for your valuable and insightful comments.

Indeed, being a structurally simple yet effective powering mechanism, surface tension gradient-induced propulsion mechanism is well-suited for small-sized soft robots propelling on water in an untethered manner. However, the lack of effective control strategies for rapid on-demand local tension tuning mechanisms and hydrodynamic interventions results in real-time dynamic power supply management and instant braking being rather difficult, leading to clumsy interactive motions in dynamic scenarios. It manifests as uncontrollable instantaneous kinematics, e.g., lacking rapid, on-demand, real-time multiple halt/start maneuverability, making it challenging to achieve both agile motion control and comparable comprehensive kinetic performance as that of natural insects. This work is to tackling the challenges of on-demand, high-performance propulsion and maneuverability, via systematic bio-inspiration from the multiscale structures and motion behavior.

Building upon your invaluable feedback, we have revised the introductory section to provide a clearer delineation of the tackling issue and the existing challenges, making it more accessible to a wider audience.

Revisions in the manuscript

[Main text: As a structurally simple yet effective propulsion mechanism, surface tension gradient-induced propulsion is well-suited for small-sized soft robots propelling on water surfaces in an untethered manner^{15,16}. However, lacking effective control strategies for a rapid, on-demand local tension tuning ability and hydrodynamic intervention limits its enormous potential to be applicable in robot manipulation. Particularly, real-time dynamic power supply management and instant braking are rather difficult, leading to very limited manoeuvrability in dynamic scenarios. It manifests as uncontrollable instantaneous kinematics, e.g., a lack of rapid, on-demand, real-time multiple halt/start manoeuvrability, making it challenging to achieve both agile motion control and comparable comprehensive kinetic performance as that of natural insects.]

Comment 3.

It would be better to clarify lessons from this robot or a new enabling technology for this robot.

Response to Comment 3. We appreciate your valuable and insightful comments. Following the completion of this research, we have undeniably acquired a wealth knowledge and insights from this robot. These findings have not only provided valuables lessons but have also paved the way for the development of new enabling technologies in the field.

Lessons:

From an engineering perspective: the natural world boasts an array of elegantly evolved structures that have developed over countless years. These structures serve as an exceptional source of inspiration for robotics research. Through systematic exploration of these biological structures and principles, we can uncover novel design concepts and enabling biological or over-biological artificial robots. This systematic bio-inspiration strategy proves to be a robust engineering approach for enabling the design of artificial robotic systems.

From a scientific perspective: exceptional performance is intricately linked to the underlying scientific principles. By delving into these principles, we can gain a deeper understanding and interpretation of certain unique phenomena. This scientific theoretical explanation contributes to the development of knowledge boundaries and has the potential for translation to other similar systems, producing a significant impact.

New enabling technologies

The on-demand fuel delivery (On-demand surface tension tuning) and posture changing-induced braking mechanisms developed in this work can be served as enabling technologies, extended to many other similar system if well designed. This can open up the application potential of utilizing surface tension gradient to propel small-scale skimming robots on water surface as well as to achieve braking.

Following your insightful suggestions, we revised our manuscript accordingly by adding new discussions regarding the above several aspects.

Revisions in the manuscript

[Main text: This work also hints that through systematic exploration of these elegantly evolved biological structures and principles, we can uncover novel design concepts and enable biological or over-biological artificial robots. We can gain a deeper understanding and interpretation by delving into these principles and the science underneath such phenomena. Indeed, the idea and principle of the on-demand fuel delivery and momentum transfer braking mechanism can be applied or translated to various other similar water-skimming robots. The potential of achieving these outcomes is not confined to soft materials alone; it can also be realized by employing well-designed rigid, jointed mechanisms as well as soft and flexible mechanisms. The key to making such a system work is the introduction of triggerable smart artificial muscles for power management and momentum transfer-based braking mechanisms.]

Comment 4.

Please reduce rhetoric words, and keep objective tone all over the manuscript.

Response to Comment 4. We genuinely value your valuable suggestions. In order to render the descriptions in the manuscript more objective and precise, we have conducted a comprehensive revision of the main text. Furthermore, we have sought assistance from Nature Research Editing Service to refine the language, resulting in a marked enhancement of the overall writing quality. These efforts have been made in response to your insights, and we believe they have greatly improved the manuscript.

Once again, we sincerely appreciate your insightful comments. As explained in the above point-by-point response, we firmly believe that our work is replete with new scientific findings, adequate theoretical explanations, and groundbreaking performance achievements. We consider this systematically bio-inspired soft propulsor to be a significant benchmark in the field, capable of inspiring other notable works. Furthermore, our work aligns well with the special collection on “Soft Robotics: Sensing, Actuation, and Integration” in this journal, particularly the topics of “Fluid-driven/magnetic-driven soft robot and Soft swimmer”.

Finally, we extend our heartfelt gratitude for your valuable recognition, support, and suggestions regarding our work. With meticulous reference to your insightful input, we have meticulously revised our manuscript, leading to a substantial enhancement in the overall quality of this work. Given your high-level perspective guidance on our work and our careful revision, we are confident that this work is well-suited for publication and poised to make a significant impact in this field.

Reviewer #3

General Comments. The authors have developed the Uni-SoPro, a Marangoni effect microskimmer that shows exceptionally high relative accelerations, speed, and decelerations. The Uni-SoPro is inspired by the rove beetle that dispenses a surfactant from its tail when dipped in the water (often to escape predators). Overall, the paper presents a lot of data and I find the fiber dispensing and braking interesting. I think the paper could be significantly improved through a more thorough discussion of motivation and methods, as well as a discussion of how these results could translate to other systems.

Response to General Comments. We sincerely appreciate the reviewer's thorough evaluation, acknowledgement and encouragement of our work. By following your valuable advice, we have carefully revised the manuscript, which results in substantial improvements in the presentation of our results. Your guidance and insights have played a pivotal role in elevating the overall quality of our work.

For the ease of tracking, the corresponding modifications in our manuscript are marked in "blue".

Comment 1.

Many of the robots/organisms that the authors compare to in Fig 1, Table S1, and the introduction to motivate their work are land-based and therefore not a very good point of comparison given the different physics involved. I'm not familiar with the term 'comprehensive kinematic performance index' (I didn't find this with google so maybe it is defined by the authors?), but if you really want to compare to organisms with higher relative accelerations and velocities, check out LaMSA organisms (e.g., [\[https://www.science.org/doi/full/10.1126/science.aao1082\]](https://www.science.org/doi/full/10.1126/science.aao1082) — there is a table in this paper). In addition, there are several recent papers (including review papers) that cover many of the same kinds of ideas for liquid-based small-scale robots including reconfigurability, systems that utilize the Marangoni effect in a controllable manner, and micro jets.

* [\[https://pubs.rsc.org/en/content/articlehtml/2023/sm/d2sm01468h\]](https://pubs.rsc.org/en/content/articlehtml/2023/sm/d2sm01468h)(DOI: 10.1039/D2SM01468H)

* Review paper: [\[https://onlinelibrary.wiley.com/doi/full/10.1002/adma.202205732\]](https://onlinelibrary.wiley.com/doi/full/10.1002/adma.202205732)

* Some of the jet-based microswimmers are a better point of comparison and should certainly be discussed. Medina-Sánchez, Mariana ; Magdanz, Veronika ; Guix, Maria ; Fomin, Vladimir M. ; Schmidt, Oliver G. Advanced functional materials, 2018, Vol.28 (25), p.1707228

It should be noted that in many of the chemically powered options, an important metric is how long they can be operated for. I don't recall seeing this metric in the authors' work.

Response to Comment 1. Thank you for providing so many valuable literatures. Following your suggestions, all the available and appropriate data that contains both necessary speeds, (de)acceleration information are supplemented and compared. In particular, we supplemented a new category called "small-scale tension-

induced soft swimmers” into our comparative analysis. In this updated comparison, we not only evaluate tension-based soft swimmers with similar principles but also map out the performance distribution across various categories, thereby providing a broader and more comprehensive perspective.

In this work, the Uni-SoPros under investigation have demonstrated outstanding performance in terms of both relative speed and acceleration. To provide a comprehensive evaluation that captures their two-dimensional performance, we have defined a Comprehensive Kinematic Performance Index (CKPI). CKPI is a unified benchmark, defined (by the authors) as a product of relative peak acceleration, BL/s^2 , and relative peak velocity, BL/s , where relative performance refers to the ratio of performance to body length. This index allows for the simultaneous assessment of relative velocity and relative acceleration in both natural organisms and robots, ensuring a holistic evaluation of their kinematic capabilities.

Furthermore, based on your recommendations, we carried out duration experiments on the Uni-SoPros. We employed a Uni-SoPro with a scale of 1.2, enabling it to execute reciprocal motion within a water tank measuring approximately $18\text{ cm} \times 28\text{ cm} \times 3.5\text{ cm}$. It exhibited a runtime of 19.4 minutes, covering a cumulative distance exceeding 110 meters. It’s worth noting that this duration experiment took place in a confined water environment without water renewal. In a more open water setting, its endurance capacity is theoretically even greater. However, it’s important to highlight that our primary focus in this work wasn’t solely on endurance. Thus, we haven’t deeply into this aspect.

In addition, relevant technologies suggested by you have been also cited and discussed within the manuscript. The endurance metric has been provided in the manuscript. Your insights have been invaluable in enhancing the comprehensiveness and relevance of our research.

Revisions in the manuscript

[Main text: Certainly, the landscape of small-scale robotics is adorned with a plethora of elegant technologies that empower these diminutive marvels. Notable examples include the catalytic artificial muscle-based insect-scale robot¹⁰, magnetic millirobot¹¹, jet-based microswimmers¹², Marangoni effect-based microbots¹³, and a myriad of others¹². These small-scale enabling technologies have significantly expanded the repertoire of motion forms and capabilities attainable by small-scale robots. Despite intensive efforts to develop untethered, insect-scale soft robots, their comprehensive kinematic performance (speed, acceleration, and deceleration) and on-demand motion behaviour still cannot be comparable/superior to their natural counterparts^{10,14}.

Main Figure:

Supplementary Table:

Table S1. Comparison of comprehensive kinematic performance.

Category	Subclass	Body length (m)	Peak velocity (m/s)	Peak acceleration (m/s ²)	Peak relative velocity (BL/s)	Peak relative acceleration (BL/s ²)	CKPI (BL ² /s ³)	
This work	Uni-SoPro	0.0036	~0.72	~30	~200	~8000	~1.6×10 ⁶	
Tension-induced swimmers	Micro swimmers (19)	0.012	0.05	0.19	4.17	15.83	6.6×10¹	
	Untethered fully soft robot (13)	0.045	0.24	0.29	5.50	6.60	3.6×10 ¹	
Mammal	Cheetah (20)	1.0500	19.900	8.300	18.95	7.90	1.5×10²	
	Impala (20)	1.3200	13.800	5.700	10.45	4.32	4.5×10 ¹	
	Lion (20)	1.7600	13.900	5.200	7.90	2.95	2.3×10 ¹	
	Zebra (20)	2.1400	10.600	3.900	4.95	1.82	9.0×10 ⁰	
	Rats (21)	0.2000	1.250	1.030	6.25	5.15	3.2×10 ¹	
	Bolt (22)	1.9500	12.340	5.659	6.33	2.90	1.8×10 ¹	
Arthropods	Dragonfly (23)	0.0557	3.400	25.000	61.04	448.83	2.7×10 ⁴	
	Flat spider (24)	0.0095	0.600	43.600	63.16	4589.47	2.9×10 ⁵	
	Grasshopper (25)	0.0750	3.210	220.960	42.80	2946.13	1.3×10 ⁵	
	Mosquito (26)	0.0040	0.400	17.590	100.00	4397.50	4.4×10⁵	
Fish	Water strider (27)	0.0118	0.991	11.773	84.00	997.71	8.4×10 ⁴	
	Bass (28)	0.2360	2.500	110.000	10.59	466.10	4.9×10 ³	
	Goldfish (29)	0.0850	2.899	247.735	34.10	2914.53	9.9×10 ⁴	
	Catfish (29)	0.0920	2.714	271.400	29.50	2950.00	8.7×10 ⁴	
	Garfish (29)	0.1200	2.364	189.120	19.70	1576.00	3.1×10 ⁴	
Soft robots	Common hatchetfish (29)	0.0210	1.407	140.700	67.00	6700.00	4.5×10⁵	
	Spiny eel (29)	0.2550	2.175	185.910	8.53	729.06	6.2×10 ³	
	GoQBot (30)	0.1000	0.750	73.397	7.50	733.97	5.5×10 ⁴	
	Fish robot (31)	0.3400	0.190	0.220	0.56	0.65	3.6×10 ⁻¹	
	Small legged robot (32)	0.0700	0.183	1.970	2.61	28.14	7.3×10 ¹	
	Explosion jump robot (33)	0.0800	3.430	114.33	42.88	1429.13	6.1×10⁴	
	Wormlike robot (34)	0.2000	0.041	0.296	0.21	1.48	3.1×10 ⁻¹	
	ISTAR (35)	0.1200	0.350	3.920	2.92	32.67	9.5×10 ¹	
	Rigid robots	DC motor (36)	0.1040	4.900	28.170	47.12	270.87	1.3×10 ⁴
		Sprawlitia (37)	0.1600	0.625	8.580	3.91	53.63	2.1×10 ²
Climbing robot (38)		0.4000	0.560	5.400	1.40	13.50	1.9×10 ¹	
Jumping strider (39)		0.0980	1.670	207.783	17.04	2120.23	3.6×10 ⁴	
Locust-inspired robot (40)		0.14	9	400	64.29	2857.14	1.8×10⁵	

Comprehensive kinematic performance index (CKPI, BL²/s³=peak relative acceleration, BL/s² × peak relative velocity, BL/s) is introduced to simultaneously evaluate the two-dimensional performance (relative velocity and relative acceleration) of the natural livings and robots. Some kinematic performance values are estimated or calculated from figures, data or videos reported.

Supplementary Note 3: In addition, we carried out duration experiments on the Uni-SoPros. We employed a Uni-SoPro with a scale of 1.2, enabling it to execute reciprocal motion within a water tank measuring approximately 18 cm × 28 cm × 3.5 cm. Remarkably, it exhibited a runtime of 19.4 minutes, covering a cumulative distance exceeding 110 meters. It's worth noting that this duration experiment took place in a confined water environment without water renewal. In a more open water setting, its endurance capacity is theoretically even greater.]

Comment 2.

How was kinematic performance measured and what is the uncertainty in these measurements? It looks like some of the measurements were taken at 2000 fps but this is not clear in the kinematic characterizations section (different from kinetic which often includes dynamics btw). I'm not familiar with and couldn't find any information on the Phantom R1212 camera. The Phantom V1212 camera is capable of full resolution at much higher frame rates (over 12000 fps) which would provide a much more accurate measurement of acceleration. Similarly, what is the lens used on this camera? Any magnification? It would be useful to know the positional error. Similarly, how many trials were used to determine these measurements?

Response to Comment 2. Thank you for your valuable and professional comments. Specifically, all the kinematic characterizations were conducted within a rectangular plastic container (18 cm × 28 cm × 3.5 cm). The container was filled with deionized water (~2 cm depth) at a temperature of approximately 20°C. The container was centrally positioned within a 3D Helmholtz coil (3CHY20-100, CH-Magnetolectricity Technology Co., Ltd) that generated a magnetic field. The strength or direction of this field were controlled either by a computer program or manually via a handle. To capture the Uni-SoPros' motion, a high-speed camera was positioned above the contained to record videos. Subsequently, the speed and acceleration of the Uni-SoPros were extracted from these recorded videos using MATLAB. The primary uncertainty in these measurements mainly come from the consistency of the assembly Uni-SoPros and potential noise (e.g, light noise, which can affect the subsequent data extraction of speed and acceleration values).

The kinematic performance/demonstrations in Fig. 1f and g, Fig. 2h, Fig. 3b-e, Fig. 4b-d, Fig. 6 and Supplementary Fig. 2-5, 7, 9, 13 and 14, 16, 18 and 19 were all recorded by the high-speed camera at 2000 fps. More detailed measurement setups are supplemented in the section of kinematic characterizations.

The reviewer must be an expert of high-speed cameras, the used camera indeed can work at 12600 fps (and even higher). However, limited by the magnetic platform, we have to provide a sufficient duration of high-speed camera for operating the magnetic platform and wait for its response. Therefore, the selection of a 2000 fps recording motions with a duration of 24.83 seconds was deliberate, as it provided an ample duration and temporal resolution (time resolution: 0.0005 s/frame, which means the position resolution was approximately 0.36 mm per frame for the observed maximum speed and the angle resolution was 2.3 degrees

per frame for the observed maximum rotation speed) to capture the motions of the Uni-SoPros. This ample recording duration (24.83s) was necessary for us to operate the external magnetic field and its response. In contrast, the Phantom R1212 camera, which operates at 12000 fps, only provides a shorter duration of 4.138 seconds.

The high-speed camera was equipped with the AF-S Micro 60 mm f/2.8G ED lens and Sigma Macro 105 mm f/2.8 DG OS HSM. There was no extra magnification applied, and each actual length of the Uni-SoPros was calibrated using the scale within the camera's image processing program (PCC 3.6).

Considering the influence of optical noise (reflective effects from certain special angles) on the recognition the center of ROI (region of interest, Uni-SoPro) using the grayscale segmentation algorithms, based on our verification, there will be ~ 0.1 mm position error (The characteristic lengths of Uni-SoPros are from 3.6 mm to 12.6 mm, positional error is less < 5% compared with their characteristic lengths). The kinematic performance characterizations in Fig. 3d were measured three times for each point, and the error bar indicates the standard deviation.

Again, thank you for your suggestions on the measurement details, we have supplemented corresponding necessary technology details in the Method section.

Revisions in the manuscript

[Main text: Kinematic characterizations. All kinematic characterizations were carried out in a rectangular plastic container (18 cm × 28 cm × 3.5 cm), which was filled with deionized water (20°C, Milli-Q Reference, MilliporeSigma) of ~2 cm depth. The container was placed in the centre of a 3D Helmholtz coil (3CHY20-100, CH-Magnetolectricity Technology Co., Ltd) that generated a magnetic field controlled by a computer program or manual handle. The same high-speed camera at 2000 fps was used to record the kinematic motions (Fig. 1f and g, Fig. 2h, Fig. 3b-e, Fig. 4b-d, Fig. 6 and Supplementary Figs. 2-5, 7, 9, 13 and 14, 16, 18 and 19). The speed and acceleration of Uni-SoPros were then extracted through the obtained videos within MATLAB R2020b. In Fig. 3d, a series of Uni-SoPros with size scales from 0.4 to 1.4 were measured at their relative peak speed (absolute speed divided by body length, BL/s) in a nearly linear motion. The characterization magnetic field is shown in Supplementary Table 3. Each sized Uni-SoPro was prepared with three samples to calculate the average velocity and error bar. Among them, Uni-SoPros with a size scale of 0.4 was selected to measure its real-time speed, acceleration and deceleration (Figs. 3b and c). A size scale of 0.8 was selected to measure its speed during five reciprocating motions, and the speed attenuation behaviours of Uni-SoPros with/without inserted PP fibres were observed (Fig. 3e). For the above kinematic performance characterizations, the high-speed camera was equipped with the AF-S Micro 60 mm f/2.8G ED lens and Sigma Macro 105 mm f/2.8 DG OS HSM. There was no extra magnification applied, and each actual length of the Uni-SoPros was calibrated using the scale within the camera's image processing program (PCC 3.6). Considering the influence of optical noise, based on our verification, there will be ~ 0.1 mm position error

(the characteristic lengths of Uni-SoPros are from 3.6 mm to 12.6 mm, and the positional error is less than 5% compared with their characteristic lengths). In addition, for trajectory control, a 2D pattern of a dove pattern was designed in AutoCAD 2020, transformed into the program for the Helmholtz coil control program and executed by a Uni-SoPro with a size scale of 0.8. The SLR camera at 120 fps was used to record its position and then obtain its trajectory via MATLAB. For agile moving, to demonstrate the agility of Uni-SoPro, a size scale of 0.8 was selected to move in a continuously turning magnetic field under the SLR camera at 120 fps (Supplementary Movie 2).]

Comment 3.

The subsequent acceleration peaks in Fig 3B bother me — where are these coming from? Are they an artifact of the measurement or is the motion of the Uni-SoPro so variable? If you have a couple trials, can you compare them? If it exists on all of them, then why? Why does it not simply accelerate smoothly and hit a peak speed? Also, in these measurements, does it even hit its peak speed? It seems to still be going up?

Response to Comment 3. Thank you for your valuable and professional comments. The subsequent peaks mainly arise from two cases: a) The large peak corresponds to each change in velocity; b) The small peaks correspond to some noise, such as background light noise. Based on our observation, these peaks will be present in all observed performances.

Figure R16. Peak analysis in performance characterizations

For the case a), this large peak corresponds to a significant change in speed. This is caused by discontinuous acceleration during intense acceleration, with some uncertainties. We are currently unable to accurately and smoothly control this intense acceleration process accompanied by complex hydrodynamics. According to our calculations in Supplementary Table 5, the Uni-SoPro operates in an excessive state of laminar and turbulent flow ($2100 < Re < 4000$).

Supplementary Table 5. Peak speeds of the Uni-SoPros with various characteristic length and corresponding Reynold number.

Scale	Characteristic length C_{bl} (mm)	Observed peak speed v_p (mm/s)	Re
0.4	3.4~3.6	~725.7	2467~2613
0.6	5.2~5.4	~572.9	2979~3094
0.8	6.9~7.2	~244.4	1686~1760
1.0	8.6~9.0	~244.4	2102~2200
1.2	10.3~10.8	~256.9	2646~2775
1.4	12.1~12.6	~284.5	3442~3585

For the case b), this small fluctuations corresponds to a background noise-induced position recognition fluctuation. Due to the need for a strong white light source (SOLA AM 5-LCR-VA, Lumencor) for recording videos with higher contrast and brightness, there will be inevitably mirror reflection occur on the Uni-SoPro during propulsion. This may influence the grayscale distribution of the Uni-SoPro (Our algorithm extracts the contour and position of the Uni-SoPro through grayscale). Therefore, there will be a positional fluctuations/error (S_{error}) caused by the recognition of the center of ROI (region of interest, Uni-SoPro). $v_{fluctuation} = S_{error}/dt$; $a_{fluctuation} = S_{error}/dt^2$, where dt is the interval time between two keyframes used for calculation, smaller dt will lead to a bigger fluctuation with same position fluctuation. Therefore, appropriate dt and filtering algorithm will contribute to reduce such fluctuations.

The observed acceleration peaks and fluctuations induced by these two cases have also been consistently observed in other trials. This is theoretically expected and reasonable, as demonstrated in Supplementary Fig. 13. In addition, the speed shown in Fig. 3b essentially reached its maximum value towards the end of the motion (at 400 ms), subsequently exceeding the camera's field of view and the boundaries of the manipulation setup.

Supplementary Figure 13. Detailed velocity and acceleration curves of Uni-SoPros in the scale effect characterization and braking test. (a, b) Velocity and acceleration curves of a Uni-SoPro with a characteristic length of 3.6 mm in the scale effect characterizations, showing a peak relative speed of ~ 201 BL/s (725 mm/s) and peak relative acceleration of $\sim 8,372$ BL/s² (30 m/s²) in the initial acceleration state. (c, d) Velocity and acceleration curves of a Uni-SoPro with a characteristic length of 3.6 mm in the braking test, show a deceleration of $\sim -5,010$ BL/s² (18 m/s²).

To elaborate these two potential factors regarding to acceleration peak/fluctuations, we have also made supplemented explanations in the manuscript.

Revisions in the manuscript

[**Main text:** Specifically, we studied the kinetic performance of a Uni-SoPro with a characteristic size of 3.6 mm by extracting its real-time speed and acceleration (Fig. 3b). Its peak acceleration reaches ~ 30 m/s² or $\sim 8,372$ BL/s² after a short time (~ 20 ms), and the maximum kinematic speed reaches ~ 725 mm/s or ~ 202 BL/s (~ 250 ms later), as shown in Supplementary Movie 1. Particularly, during a braking test of a Uni-SoPro of the same size, as shown in Supplementary Movie 2, it exhibits braking with a deceleration of $-5,010$ BL/s², as shown in Fig. 3c, with detailed kinetic characteristics shown in Supplementary Fig. 13. During these kinematic characterizations, it is important to note that some peaks and fluctuations may appear in the acceleration curve. These variations can be attributed to potential discontinuities in the acceleration process as well as minor noise in the measurements, particularly in the presence of background light. These factors are considered possible sources of variability in the recorded acceleration data.]

Comment 4.

In a similar vein, a Helmholtz coil system has constant magnetic field in the center, but a gradient towards the edges. From the movie, I could not tell exactly how far the Uni-SoPro was traveling in each experiment and what this means for the field gradient that is present towards the edges of the coil system. For example, the X and Y positions in Fig 3F seem to extend to the edge of the coils. Was there a specific goal trajectory in this figure?

Response to Comment 4. Thank you for your valuable and professional comments. To ensure a sufficiently large uniform magnetic field space, in this work, we used a sufficiently large Helmholtz coil platform ($\varphi_x = 834$ mm; $\varphi_y = 633$ mm; $\varphi_z = 455$ mm), as shown in Figure R17.

Figure R17. Manipulation platform

All the characterization experiments in this work are operated in the center of the 3D Helmholtz coils that can generate a up to 10 mT uniform magnetic field in the central spherical volume with a radius of 75 mm, uniformity ~95%. The unilateral strokes of Uni-SoPros were less 150 mm and within in such a nearly magnetically uniform spherical space in these characterizations. The demonstration in Fig. 3f is essentially operated within this range. Additionally, it's worth noting that in this preliminary demonstration, we did not employ the standard model parameters for manipulation. Therefore, we could not obtain a theoretical trajectory based on the developed model. Instead, we have placed this comparison in the subsequent modeling section, as shown in Fig. 6c and d. This approach allowed us to provide a more comprehensive analysis of the model's performance and its alignment with the observed behavior.

Fig. 6. Kinematic modeling for multi-segment motion trajectory planning. **a** Multiple movement programming can be further decomposed into corresponding subsegments. Each subsegment can be divided into four typical stages: acceleration, stable propulsion, deceleration and in-situ swerving, respectively. **b** According to the above typical four motion stage, a general motion model was proposed for movement programming. With the established inverse solution model, critical manipulation parameters of each subsegment can be reversely solved. **c** Experimental and model velocity comparison of with/without fiber inserted Uni-SoPros (~7.2 mm). **d** 10 consecutive cycle kinematic curve comparisons of the predicted one and the experimental one and the corresponding error analysis.

To make the experiment conditions clearer, we supplemented descriptions in the Supplementary Fig. 1.

[Supplementary Figure:

Supplementary Figure 1. Manipulation platform and operation schematics of Uni-SoPros. (a) A manipulation system consists of Helmholtz coils (up to 10 mT uniform magnetic field in the central spherical volume with a radius of 75 mm, uniformity ~95%), a high-speed imaging system, and an operation platform. All the kinematic characterizations were operated in such a magnetically uniform spherical area. **(b)** Schematic of Uni-SoPro actuation on water surface. **(c)** Two operational manners for manipulation of Uni-SoPros. Automatic control manner is used for pre-programmed actuation while manual control is used for real-time control.]

Comment 5.

I find the addition of fibers into the tail intriguing — this seems to be an interesting point. While Fig 2F discusses the tuning capability, I would be interested in seeing how this relates to the incredible accelerations and decelerations seen in the final system. Speaking of which, I’m not sure how tuning capability is defined in the text or in Fig 2F.

Response to Comment 5. Thank you for your valuable comments. Thank you for your positive comments that the addition of fibers into tail is intriguing, which was proved to prevent the fast attenuation of the speed induced by the excessive and rapid release of surfactants.

Indeed. The local surface tension tuning capability (abbreviated as STTC) is an ability to decrease the surface tension behind the Uni-SoPro. As shown in the following equation, larger STTC means will induce a larger tension gradient-induced and momentum induced propulsive force, therefore, it is a critical for evaluating the propulsion capability (Notice that if STTC was approaching 0, the Uni-SoPro would not be pushed forward).

$$F = \int \underbrace{\rho_f g h(x) l(x) \sin \beta dx}_{\text{hydrostatic}} + \underbrace{\mu_f S \frac{dv}{dz} \cos \beta}_{\text{viscosity}} - \underbrace{\int_c \gamma t \frac{v}{|v|} dl}_{\text{tension}} + \underbrace{R_w}_{\text{wave resistance}} - \underbrace{\rho_f S_1 (v_c - v)^2 \sin \beta}_{\text{momentum}}. \quad (1)$$

We defined and introduced this index to facilitate the comparison of propulsion attenuation effect during multiple fuel deliveries with or without of PP fiber into its tail, Fig. 2e and f. Through our experiments, we found that when such a fiber is inserted, it can provide a more stable STTC during multiple fuel deliveries. A high STTC metric is indeed correlated with the incredible accelerations observed, while a stable STTC metric reflects the consistency of performance during multiple fuel deliveries. This is subsequently substantiated by the speed attenuation comparison in Fig. 3e.

Fig. 3e. Speed attenuation comparison between a 7.2 mm Uni-SoPro with/without fiber in 10 one-way cycles. The speed of that with an inserted fiber attenuates significantly slower than that without fiber.

To elaborate the metric of STTC clearer, we revised and supplemented descriptions in the Manuscript.

Revisions in the manuscript

[Main text: By measuring the surface tension decrement after every ten triggers from the first ten times (10^{1st}) to the sixth ten times (10^{6th}) on freshwater surfaces, we studied their local surface tension tuning capabilities (STTC) with the tails with/without fibres (Fig. 2f and Supplementary Fig. 10). There was a dramatically degressive local tension tuning capability as the trigger number increased when the fibres was not inserted. By contrast, those with fibres show a slow and linear-like attenuation of local tension tuning, particularly at a 200-ms trigger time. The attainment of a sufficient and stable STTC metric is of critical importance, as it directly impacts the overall kinematic performance and its consistency during multiple triggering. Additionally, we found that by controlling the dipping duration, it is easy to control the dipping

behaviour by magnetic fields. Such precise tension tuning, which embodies energy supply regulation, offers a relatively ideal trigger attribute by providing a sufficient time window with expectable velocity for precise manipulation, and further trajectory control/planning.]

Comment 6.

I'm not sure how the measurement in Fig 3E was done? Was the UniSoPro turned around each time? Was the amount of surfactant consistent between trials? Is the 'without fiber' decreasing because it releases most of its surfactant at the beginning?

Response to Comment 6. Thank you for your valuable comments. In this comparison experiment, two UniSoPros (7.2 mm) with and without fiber inserted were compared to propel 10 one-way cycles. Due to the limited manipulation space, the Uni-SoPro would turn around each time and run back and forth in the magnetically uniform zone. The total amount in the fuel tank is consistent, but the presence and absence of fibers affect the fuel release. As you said, without inserted fiber, there will be more fuel dispensing in the first few times compared the later ones, while the fuel delivery is more consistent (more stable STTC) with fiber inserted.

Additionally, we conducted a discussion on these two fuel dispensing strategies through numerical simulation, as illustrated in Supplementary Figure 17 and Note 4. Our findings indicate that the strategy of the gradual decrease in the release of fuel (without fiber inserted) initially yields a substantial theoretical propulsive force. However, it experiences a faster decay compared to the more constant strategy (with fiber inserted).

Supplementary Figure 17. Comparison of surfactant release with different strategies via numerical simulation.

We appreciate your feedback, and to enhance the clarity of the experiments, we have included additional descriptions in the manuscript.

Revisions in the manuscript

[Main text: Since a gentler delivery strategy discussed in the previous section can significantly slow the velocity attenuation and hence expectable trajectory for longer-term propulsion, two Uni-SoPros with/without fibres inserted were compared to propel 10 one-way cycles subjected to the same magnetic triggering conditions (Fig. 3e). Due to limited space, Uni-SoPros turn around each time. We observed a rapid decline in velocity for that without fibres, whereas the Uni-SoPro with fibres demonstrated a nearly constant trend during the test window. This result suggests that the inserted fibres, which mimic the structure and mechanism of conducting canals in the beetle gland system, provide much more precise fluidic regulation/delivery. Consequently, it minimizes velocity decline and simplifies later trajectory planning for multiple triggers. Moreover, a complementary numerical simulation also suggests that even released surfactant during multiple triggers can provide a more stable tension gradient for continuous propulsion (Supplementary Fig. 17 and Supplementary Note 4).]

Comment 7.

I found the dynamic scenario somewhat challenging to understand. While the text and Fig 7 indicates that the Uni-SoPro is responding to these events, I assume that it is either programmed into the triggering of the Helmholtz coil or a person is triggering the different behaviors. As such, everything is pre-programmed in this scenario. If I got this wrong, then much more discussion of the feedback mechanism is needed for clarity. If this is a correct interpretation, then I think the way it's being told is overselling the situation and can be very confusing for a reader.

Response to Comment 7. Thank you for your valuable comments. The authors would like to appreciate the reviewer for raising the concern, which we believe is caused by misunderstandings that mainly suggest we need to improve our presentation. Here, we want to demonstrate the Uni-SoPro's pre-programmable capacities (without real-time responding) in a complex scenarios with spatial constraints and temporal constraints (in an open-loop manner, only with good braking and motion modeling/programming abilities can discrete preprogramming be done well). Therefore, we set a dynamic labyrinth outfitted with preprogrammed time-varying signal lights that will periodically change their states every 2 s. When the light is off, the Uni-SoPro requires passing through the labyrinth as quickly as possible without hitting the walls from the start point to the endpoint in an open-loop control manner.

Such a setup makes precise/instantaneous halts necessary where/when the light is on, and instantaneous starts necessary when the light is off. Such a setup makes precise/instantaneous halts necessary where/when the light is on, and instantaneous starts necessary when the light is off. Based on the above spatiotemporal constraints, the accumulated stroke error $\Delta u_{\text{overall}}$ should be within 14 mm and $\Delta u_{1,6,16}$ should be smaller than -5.5 mm during the whole passing process (Supplementary Fig. 18 and Supplementary Note 7). For the

demonstrated Uni-SoPro, it only has ~0.25 s to halt for each turning, otherwise, it will miss the best-turning window. In addition, if the braking behavior was poor (braking distance large than 6 cm), the Uni-SoPro could not pass such a labyrinth.

Despite these challenges, the demonstrated Uni-SoPro successfully halted and crossed each traffic signal, turning intersection without any wall strikes during the process, and finally reached the endpoint within 20 s, Fig. 7c and Supplementary Movie 6. The whole process is completed through open-loop manner, there is no any real-time feedback mechanisms. This demonstration shows an excellent manipulation even in an open-loop manner. We believe that if other closed-loop feedback mechanisms were introduced in the future, its controllability will be further enhanced. The primary focus of this work is on the systematical bio-inspired small-scale robot codesign, to enable its biological-level and even over-biological performance and controllability as well as to uncover and understand scientific mechanisms underlying these achievements. Following your valuable suggestions, to discuss this demonstration clearer and more objective, we carefully revised the corresponding descriptions in manuscript and Figure 7 as well as an objective discussion on the limitations of the current open loop control in the discussion section. We believe that the current statement will be more objective and fairer.

[Main text: To demonstrate its programmable motions, we set a labyrinth outfitted with preprogrammed time-varying signal lights that periodically change their states every 2 s (Fig. 7a). We stipulated that Uni-SoPro should not pass through the light once illuminated. When the light is off, the Uni-SoPro requires passing through the labyrinth as quickly as possible without hitting the walls from the start point to the endpoint in an open-loop control manner without any feedback mechanisms. Such a setup makes precise/instantaneous halts necessary when the light is on and instantaneous starts necessary when the light is off. Based on the above spatiotemporal constraints, the accumulated stroke error $\Delta u_{\text{overall}}$ should be within 14 mm, and $\Delta u_{1,6,16}$ should be smaller than -5.5 mm during the whole passing process (Supplementary Fig. 20 and Supplementary Note 7). The demonstrated Uni-SoPro only has ~0.25 s to halt for each turning, or else it will miss the best-turning window. Moreover, compared to that in open water, e.g., in Fig. 3f, there are some disturbances due to the influence of the wall on the tension distribution and the meniscus along the walls, and these nonlinear boundary conditions hence make precise trajectory control difficult. Therefore, there are quite a few stringent constraints for Uni-SoPro to pass through such a dynamic scenario without any feedbacks (Fig. 7b). Despite those challenges, the demonstrated Uni-SoPro successfully halted and crossed each traffic signal, turned at intersection without any wall strikes during the process, and finally reached the endpoint within 20 s in a preprogrammed manner, as shown in Fig. 7c and Supplementary Movie 7, with the key snapshots of braking, propelling, and steering illustrated in Fig. 7d and e.

There are also a few limitations in the current study. A significant limitation is that during the manipulation of Uni-SoPro in this work, an open-loop program is implemented without real-time sensing and

feedback. Thus, it does not allow for dynamic adaptation and adjustment based on the surrounding environment or changing conditions. In future studies, it would be valuable to explore the integration of a global vision system, or preferably but challengingly an energy-efficient local sensing system with edge computation to enable closed-loop manipulation of the Uni-SoPro.

Main Figure:

Fig. 7. Demonstrations of Uni-SoPro going through a dynamic labyrinth with complex spatiotemporal constraints. **a** To demonstrate its controllability, Uni-SoPro should pass through such a 7×10 water labyrinth as soon as possible, following the following defined rules: a) In-situ stationary swerving at each

intersection/corner. **b**) **Obeying** the signal traffic light: it can only pass through when the frontal signal light is off and should brake in front of the light when it is on. **b** Strict spatial and temporal constraints during passing through such a dynamic labyrinth. **c** A Uni-SoPro crosses through a labyrinth with time-varying signal lights. **d** Close-up snapshots of Uni-SoPro when it brakes and goes through one of the signal lights. **e** Close-up snapshots of Uni-SoPro when it steers at turning.]

Comment 8.

The authors state that a magnetic tail with a ‘longer length, finer caliber, and lower magnetic particle count exhibited a more favorable deflection response...’ — There is really nothing in Figs B-D that is at all surprising and if trying to condense, this could be cut. There is no real optimal found here — just expected trends.

Response to Comment 8. Thank you for your valuable comments and feedback. The previous characterizations may provide limited guidance for the design of the magnetic tail. Actually, the idea deflection response of magnetic tail is crucial for the on-demand motion manipulation. On the one hand, the magnetic tail should bend downward for contacting the water surface and ensuring effective fuel delivery. On the other hand, the magnetic tail should bend upward for cutting off the fuel supply on demand by overcoming the solid-liquid interface energy. Therefore, on the limited operation magnetic field, the suitable magnetic tubes are crucial. To enhance this characterizations, we additionally conducted the fuel delivery and detaching experiments with various magnetic tail, as shown in Fig. R18. Our findings revealed that a magnetic tail with higher magnetic particle contents exhibited more favorable bending behaviors for ensuring effective fuel delivery to water surface and on-demand cutting off fuel supply by overcoming the solid-liquid interface energy. In this characterization, we found that the tail with too low concentration of magnetic particles would cause fuel delivery (contacting water surface) and desorption (cut off fuel delivery) process failure. In this work, to ensure robustness, the higher concentration (PDMS: MPs = 21: 40) was adopted for the experiments.

Figure R18. Characterizations of magnetic tails. Effect of mixture ratio of PDMS and magnetic particles (MPs), magnetic tail size scale on the bending response under various external applied magnetic field B_z . Error bars indicate the standard deviation for $n=3$ sample measurements at each data point. **d** Detaching

behavior of the magnetic tail from the water surface with different mixture ratio of PDMS and magnetic particles.

We sincerely appreciate your valuable comments and feedback. Based on the new results obtained, we have incorporated a comprehensive discussion into the relevant sections of the original text. These additional characterization results and discussions further enhance the overall quality and completeness of our manuscript.

Revisions in the manuscript

[Main text: We examined the influence of the magnetic particle content and magnetic tail size scale on its behaviours of downward bending for fuel delivery and upward bending for detaching from the water surfaces, as shown in Fig. 2b-d. Our findings revealed that a magnetic tail with higher magnetic particle contents exhibited more favourable bending behaviours for ensuring effective fuel delivery to the water surfaces and on-demand cutting of the fuel supply by overcoming the solid-liquid interface energy. In addition, we also characterized the influence of different postures and submerged magnetic tail lengths on swerving behaviour (Supplementary Fig. 7). The results show that excessive immersion of the magnetic tail in water will attenuate its swerving behaviour. Thus, selecting a suitable magnetized tail with appropriate magnetic particle contents, tail size scale and submerged length of tail is very significant for achieving efficient manoeuvrability.

Main Figure:

Fig. 2. Systematic bio-inspiration from its natural counterparts (*Stenus comma*), and codesign of the soft propulsor with detailed characterizations. a Systematic bio-inspiration designs from macro to micro as well as artificial codesign to enable the presented soft propulsor with a triggerable tail, stable fuel transport and delivery, nonwetting surfaces, and decoupled steerable capability. **b-c** Effects of the mixture ratio of PDMS and magnetic particles (MPs) and magnetic tail size scale on the bending response under various external applied magnetic fields B_z . The error bars indicate the standard deviation for $n=3$ sample measurements at each data point. **d** Detaching behaviour of the magnetic tail from the water surface with different mixture ratios of PDMS and magnetic particles.]

Comment 9.

The discussion section could use some further discussion of how these results could be translated to other systems. For example, could the braking ideas be useful for the numerous other means for controlling the Marangoni effect discussed in the papers cited earlier in this review? Is this only useful in a Helmholtz coil system or could it be useful in some more autonomous water skimming robots at some point?

Response to Comment 9. We appreciate the reviewer's suggestion to discuss the broader implications of our findings and their potential applicability to other systems. Indeed, the principles behind the braking mechanisms we've explored in this study hold promise for various applications beyond our specific experimental setup. Theoretically, the braking idea by transmitting the kinematic energy to the surrounding water or external environment as unveiled in this study, holds the potential to be effective for the numerous other means for controlling the Marangoni effect, e.g., to introduce an on-demand, triggerable braking mechanism to interact with the aquatic environment.

In addition, this on-demand fuel delivery and braking mechanism can also be translated to other autonomous water skimming robot, for instances, the magnetic tail can be replaced by other smart artificial muscle to enable on-demand power management. In addition, the braking behavior can be realized similarly by introducing an external triggerable mechanism to influence its hydrodynamics.

Thank you for your valuable and insightful comments. We have supplemented corresponding discussions in the DISCUSSION section.

Revisions in the manuscript

[Main text: Indeed, the idea and principle of the on-demand fuel delivery and momentum transfer braking mechanism can be applied or translated to various other similar water-skimming robots. The potential of achieving these outcomes is not confined to soft materials alone; it can also be realized by employing well-designed rigid, jointed mechanisms as well as soft and flexible mechanisms. The key to making such a system

work is the introduction of triggerable smart artificial muscles for power management and momentum transfer-based braking mechanisms.]

Comment 10.

There is significant emphasis on being soft, especially in the abstract and introduction, but it is unclear what the importance of this is besides reconfiguration (which can be done in both more rigid, jointed mechanisms as well as soft mechanisms)

Response to Comment 10. We would like to extend our sincere gratitude for your valuable and professional comments. We share your perspective that many of the structures and functions we've explored in this study could potentially be achieved using both rigid, jointed mechanisms and soft/flexible mechanisms, provided they are appropriately designed.

In this work, we've taken a deliberate approach to facilitate the implementation of a systematic, bio-inspired artificial robot. To achieve this, we've harnessed the capabilities of function particle-doped soft materials. Specifically, we've used materials doped with magnetic particles and carbon black to create the necessary components, such as the magnetically on-demand bendable tail for fuel delivery and cutoff, the superhydrophobic skin with microstructure, fuel refinement fibers for smoothing fuel delivery, and physical steering chips.

These soft materials, cured from a prepolymer base, offer us a versatile foundation for tailoring various aspects of our propulsor. They provide us with the means to control magnetism, facilitate laser surface processing, achieve the appropriate stiffness for large deformations, and manage overall density.

By choosing these materials, we've aimed to strike a balance between flexibility and functionality, enabling us to achieve the desired outcomes in our study. Thank you again for your valuable insights and considerations. To explain this clear, we added necessary explanations in the manuscript.

Revisions in the manuscript

[Main text: Indeed, the idea and principle of the on-demand fuel delivery and momentum transfer braking mechanism can be applied or translated to various other similar water-skimming robots. The potential of achieving these outcomes is not confined to soft materials alone; it can also be realized by employing well-designed rigid, jointed mechanisms as well as soft and flexible mechanisms. The key to making such a system work is the introduction of triggerable smart artificial muscles for power management and momentum transfer-based braking mechanisms.]

Comment 11.

Why is the speed not = 0 after braking in Movie S2?

Response to Comment 11. We greatly appreciate your valuable comments. As revealed by our comprehensive CFD analysis, it is evident that the braking process is highly effective in transferring a significant portion of the kinetic energy (approximately 92.5%) from the Uni-SoPro to the surrounding water. However, as is often the case, some residual kinetic energy remains even after the braking process.

Furthermore, it's important to note that the surface tension gradient doesn't dissipate instantaneously, and this lingering gradient contributes to a certain rebound in speed. Notably, we have observed that both this residual velocity and the rebound effect are influenced by the initial cruising posture of the Uni-SoPro and the dynamics of its posture changes, as exemplified in Fig. 4b-d.

In addition, this braking behavior can be finely tuned through the manipulation of magnetic fields, according to the motion demands.

Fig. 4. Braking hydrodynamics analysis. **a** Schematic diagram of the braking process: (S1) Steady cruising state under the actuation of surface tension gradient and momentum transfer of Marangoni flow to the caudal fin of the Uni-SoPro; (S2) In the triggering state, the magnetic tail is cocked up by the external magnetic field to terminate fuel supply and the motive force; (S3) Immediately, subjected to the magnetic torque from the magnetic tail and steering chips, the induced fast transformation of the Uni-SoPro from the tilt-back state to tilt-forward state triggers a drastic momentum transfer to the water, where this momentum transfer greatly

attenuates the kinetic energy of the Uni-SoPro; (S4) The Uni-SoPro tilts forward to a maximum state and subsequently returns to a stable state under the combined action of the magnetic torque and surface tension. **b** Experimental verification of the whole braking process under different braking magnetic fields. **c** Full cycle record of the position and attitude of Uni-SoPro in the above experiments. The time interval of sampling points is 1.25 ms in the braking process and 6.25 ms in the cruising process. **d** Full cycle record of the velocity and attitude of the Uni-SoPros, and the results show a high relevance between the velocity and attitude. **e** The changing of subjected forces in different states (from state S1 to state S4).

To provide a lucid elucidation of these findings, we have thoughtfully incorporated essential clarifications within the manuscript.

Revisions in the manuscript

[Main text: To gain deeper insights into the braking dynamics, we conducted a series of gradient braking experiments. These experiments involved the application of magnetic fields with identical magnitudes (100 Gs), albeit with varying angles (defined as the included angles of Magnetic Field \mathbf{B} and z -axis, denoted as φ , and here, 10° , 45° , and 60° were selected), as shown in Fig. 4b. As observed, different included angles of the magnetic fields lead to different posture changes and trajectories (Fig. 4c). By extracting key parameters (the velocity and titling angle β) of Uni-SoPros during the process, as shown in Fig. 4d, we observed that the velocity attenuation and subsequent rebounding are highly related to the external magnetic field.]

Comment 12.

For scaling, were these geometrically scaled? What dictated the shape of the Uni-SoPro?

Response to Comment 12. We extend our heartfelt gratitude for your invaluable and professional comments. Regarding the scaled Uni-SoPros in our study, we primarily applied geometric scaling to magnify them. However, your query raises an important consideration. To optimize their performance at varying scales, additional design modifications might indeed be necessary. Your discerning question underscores the significance of not only scaling but also tailored design to ensure the effectiveness and efficiency of these miniature robotics systems.

Once again, thank you for your thoughtful input, which encourages us to further refine our research.

In addition, to discuss this clearer, we added the necessary discussion in the manuscript.

Revisions in the manuscript

[Main text: To learn the scale effect on the kinematic characteristics of Uni-SoPros, we investigated a series of Uni-SoPros that are geometrically scaled with various characteristic sizes on movements via a high-speed recording system. The results show a clear trend where the peak velocity monotonically decreases with

increasing size, particularly at smaller scales, as shown in Fig. 3d, with detailed structural parameters shown in Supplementary Table 2 and the flow mapping of each sample size propelled at its peak velocity shown in Supplementary Fig. 15. Dynamics analysis of basic propulsion states and performance trends related to characteristic lengths also agree well with the above findings, as shown in Supplementary Notes 2 and 3 and Supplementary Fig. 16. In addition, to optimize their performance at various scales, additional design modifications/optimization might be necessary.]

Comment 13.

I appreciate that it is challenging to write papers in English when it is not one's native language. I highly recommend using some of the AI grammar engines that exist now or a native speaker for proofing however — the paper was challenging to read. There are numerous typos throughout.

Response to Comment 13. Thank you for your suggestions. Following your advices, we have sought assistance from Nature Research Editing Service to refine the language, resulting in a marked enhancement of the overall writing quality.

Once again, we extend our heartfelt appreciation for your positive recognition and support of our work. Your valuable, insightful, and meticulous comments have been instrumental in elevating the quality of our work. We now firmly believe that, with your guidance and our diligent revisions, it is well-prepared for publication and holds the potential for the significant impact in the field of the small-scale robots.

Reference

1. R. Zhao, Y. Kim, S. A. Chester, P. Sharma, X. Zhao, Mechanics of hard-magnetic soft materials. *J. Mech. Phys. Solids*. **124**, 244–263 (2019).
2. W. Hu, G. Z. Lum, M. Mastrangeli, M. Sitti, Small-scale soft-bodied robot with multimodal locomotion. *Nature*. **554**, 81–85 (2018).
3. Y. Kim, H. Yuk, R. Zhao, S. A. Chester, X. Zhao, Printing ferromagnetic domains for untethered fast-transforming soft materials. *Nature*. **558**, 274–279 (2018).
4. H. Schildknecht, Chemical Ecology- A Chapter of Modern Natural Products Chemistry. *Angew. Chemie - Int. Ed.* **15**, 214–222 (1976).
5. J. S. Koh, E. Yang, G. P. Jung, S. P. Jung, J. H. Son, S. I. Lee, P. G. Jablonski, R. J. Wood, H. Y. Kim, K. J. Cho, Jumping on water: Surface tension-dominated jumping of water striders and robotic insects. *Science*. **349**, 517–521 (2015).
6. E. W. Hawkes, C. Xiao, R. Peloquin, C. Keeley, M. R. Begley, M. T. Pope, G. Niemeyer, Engineered jumpers overcome biological limits via work multiplication. **604** (2022), doi:10.1038/s41586-022-04606-3.
7. B. H. Kim, K. Li, J. T. Kim, Y. Park, H. Jang, X. Wang, Z. Xie, S. M. Won, H. J. Yoon, G. Lee, W. J. Jang, K. H. Lee, T. S. Chung, Y. H. Jung, S. Y. Heo, Y. Lee, J. Kim, T. Cai, Y. Kim, P. Prasopsukh, Y. Yu, X. Yu, R. Avila, H. Luan, H. Song, F. Zhu, Y. Zhao, L. Chen, S. H. Han, J. Kim, S. J. Oh, H. Lee, C. H. Lee, Y. Huang, L. P. Chamorro, Y. Zhang, J. A. Rogers, Three-dimensional electronic microfliers inspired by wind-dispersed seeds. *Nature*. **597**, 503–510 (2021).
8. G. Li, X. Chen, F. Zhou, Y. Liang, Y. Xiao, X. Cao, Z. Zhang, M. Zhang, B. Wu, S. Yin, Y. Xu, H. Fan, Z. Chen, W. Song, W. Yang, B. Pan, J. Hou, W. Zou, S. He, X. Yang, G. Mao, Z. Jia, H. Zhou, T. Li, S. Qu, Z. Xu, Z. Huang, Y. Luo, T. Xie, J. Gu, S. Zhu, W. Yang, Self-powered soft robot in the Mariana Trench. *Nature*. **591**, 66–71 (2021).
9. L. X. Lyu, F. Li, K. Wu, P. Deng, S. H. Jeong, Z. Wu, H. Ding, Bio-inspired untethered fully soft robots in liquid actuated by induced energy gradients. *Natl. Sci. Rev.* **6**, 970–981 (2019).
10. A. Pena-Francesch, J. Giltinan, M. Sitti, Multifunctional and biodegradable self-propelled protein motors. *Nat. Commun.* **10**, 3188 (2019).
11. H. Schildknecht, Chemical ecology—a chapter of modern natural products chemistry. *Angew. Chemie Int. Ed. English*. **15**, 214–222 (1976).
12. A. Pena-Francesch, J. Giltinan, M. Sitti, Multifunctional and biodegradable self-propelled protein

- motors. *Nat. Commun.* **10**, 3188 (2019).
13. L. X. Lyu, F. Li, K. Wu, P. Deng, S. H. Jeong, Z. Wu, H. Ding, Bio-inspired untethered fully soft robots in liquid actuated by induced energy gradients. *Natl. Sci. Rev.* **6**, 970–981 (2019).
 14. X. Yang, L. Chang, N. O. Pérez-Arancibia, An 88-milligram insect-scale autonomous crawling robot driven by a catalytic artificial muscle. *Sci. Robot.* **5**, 1–14 (2020).
 15. H. Lu, M. Zhang, Y. Yang, Q. Huang, T. Fukuda, Z. Wang, Y. Shen, A bioinspired multilegged soft millirobot that functions in both dry and wet conditions. *Nat. Commun.* **9** (2018), doi:10.1038/s41467-018-06491-9.
 16. M. Medina-Sánchez, V. Magdanz, M. Guix, V. M. Fomin, O. G. Schmidt, Swimming Microrobots: Soft, Reconfigurable, and Smart. *Adv. Funct. Mater.* **28** (2018), doi:10.1002/adfm.201707228.
 17. W. Yang, X. Wang, Z. Wang, Z. Yuan, Z. Ge, H. Yu, A multi-stimulus-responsive bionic fish microrobot for remote intelligent control applications. *Soft Matter.* **19**, 913–920 (2022).
 18. W. Wang, B. Han, Y. Zhang, Q. Li, Y. L. Zhang, D. D. Han, H. B. Sun, Laser-Induced Graphene Tapes as Origami and Stick-On Labels for Photothermal Manipulation via Marangoni Effect. *Adv. Funct. Mater.* **31**, 1–11 (2021).
 19. B. Wang, S. Handschuh-Wang, J. Shen, X. Zhou, Z. Guo, W. Liu, M. Pumera, L. Zhang, Small-Scale Robotics with Tailored Wettability. *Adv. Mater.* **35**, 1–28 (2023).
 20. A. M. Wilson, T. Y. Hubel, S. D. Wilshin, J. C. Lowe, M. Lorenc, O. P. Dewhirst, H. L. A. Bartlam-Brooks, R. Diack, E. Bennitt, K. A. Golabek, Biomechanics of predator–prey arms race in lion, zebra, cheetah and impala. *Nature.* **554**, 183–188 (2018).
 21. R. M. Walter, Kinematics of 90 running turns in wild mice. *J. Exp. Biol.* **206**, 1739–1749 (2003).
 22. M. Badura, Biomechanical analysis of the discus at the 2009 IAAF World Championships in athletics. *New Stud. Athl.* **25**, 23–35 (2010).
 23. S. A. Combes, D. E. Rundle, J. M. Iwasaki, J. D. Crall, Linking biomechanics and ecology through predator–prey interactions: flight performance of dragonflies and their prey. *J. Exp. Biol.* **215**, 903–913 (2012).
 24. Y. Zeng, S. Crews, Biomechanics of omnidirectional strikes in flat spiders. *J. Exp. Biol.* **221**, jeb166512 (2018).
 25. A. Koehnsen, J. Kambach, S. Büsse, Step by step and frame by frame – Workflow for efficient motion tracking of high-speed movements in animals. *Zoology.* **141**, 125800 (2020).

26. F. T. Muijres, S. W. Chang, W. G. van Veen, J. Spitzen, B. T. Biemans, M. A. R. Koehl, R. Dudley, Escaping blood-fed malaria mosquitoes minimize tactile detection without compromising on take-off speed. *J. Exp. Biol.* **220**, 3751–3762 (2017).
27. V. M. Ortega-Jimenez, L. Von Rabenau, R. Dudley, Escape jumping by three age-classes of water striders from smooth, wavy & bubbling water surfaces. *J. Exp. Biol.* **220**, 2809–2815 (2017).
28. P. W. WEBB, Speed, Acceleration and Manoeuvrability of Two Teleost Fishes. *J. Exp. Biol.* **102**, 115–122 (1983).
29. R. C. Eaton, R. A. Bombardieri, D. L. Meyer, The Mauthner-initiated startle response in teleost fish. *J. Exp. Biol.* **66**, 65–81 (1977).
30. H.-T. Lin, G. G. Leisk, B. Trimmer, GoQBot: a caterpillar-inspired soft-bodied rolling robot. *Bioinspir. Biomim.* **6**, 026007 (2011).
31. A. D. Marchese, C. D. Onal, D. Rus, Autonomous soft robotic fish capable of escape maneuvers using fluidic elastomer actuators. *Soft Robot.* **1**, 75–87 (2014).
32. H. Peng, T. Mao, X. Lu, A small legged deformable robot with multi-mode motion. *J. Intell. Mater. Syst. Struct.* **31**, 704–718 (2020).
33. M. T. Tolley, R. F. Shepherd, M. Karpelson, N. W. Bartlett, K. C. Galloway, M. Wehner, R. Nunes, G. M. Whitesides, R. J. Wood, in *2014 IEEE/RSJ International Conference on Intelligent Robots and Systems* (2014), pp. 561–566.
34. Sangok Seok, C. D. Onal, R. Wood, D. Rus, Sangbae Kim, in *2010 IEEE International Conference on Robotics and Automation* (IEEE, 2010; <http://ieeexplore.ieee.org/document/5509542/>), pp. 1228–1233.
35. D. Zarrouk, R. S. Fearing, Controlled in-plane locomotion of a hexapod using a single actuator. *IEEE Trans. Robot.* **31**, 157–167 (2015).
36. D. W. Haldane, R. S. Fearing, in *2015 IEEE International Conference on Robotics and Automation (ICRA)* (IEEE, 2015; <http://ieeexplore.ieee.org/document/7139828/>), vols. 2015-June, pp. 4539–4546.
37. J. G. Cham, S. A. Bailey, J. E. Clark, R. J. Full, M. R. Cutkosky, Fast and robust: Hexapedal robots via shape deposition manufacturing. *Int. J. Rob. Res.* **21**, 869–882 (2002).
38. D. W. Haldane, R. S. Fearing, in *Robotics: Science and Systems* (The MIT Press, 2008; <https://direct.mit.edu/books/book/2310/chapter/60363/design-of-a-bio-inspired-dynamical-vertical>), pp. 9–16.
39. J.-S. Koh, E. Yang, G.-P. Jung, S.-P. Jung, J. H. Son, S.-I. Lee, P. G. Jablonski, R. J. Wood, H.-Y.

Kim, K.-J. Cho, Jumping on water: Surface tension–dominated jumping of water striders and robotic insects. *Science*. **349**, 517–521 (2015).

40. M. Ilton, M. Saad Bhamla, X. Ma, S. M. Cox, L. L. Fitchett, Y. Kim, J. sung Koh, D. Krishnamurthy, C. Y. Kuo, F. Z. Temel, A. J. Crosby, M. Prakash, G. P. Sutton, R. J. Wood, E. Azizi, S. Bergbreiter, S. N. Patek, The principles of cascading power limits in small, fast biological and engineered systems. *Science*. **360** (2018), doi:10.1126/science.aao1082.

REVIEWERS' COMMENTS

Reviewer #1 (Remarks to the Author)

We appreciate the very thorough revision of the original manuscript and supporting material that the author have undertaken in view of the received feedback. We are impressed by the openness to feedback and the significant work undertaken to address it in a very meticulous way. This has led to a substantial revision and rewriting of the original text as well as expansion of the already considerable captions and supporting information. More importantly, the new version incorporates many new insight (including corrections of mistakes) and covers many more aspects of the work and future perspectives. Finally, the language as requestes was thoroughly revised. The manuscript is now in a form to contribute significantly to small-scale robotics and suitable for publication in the journal

Reviewer #2 (Remarks to the Author):

I thank the authors for their detailed response to my comments. In my view, the paper is of sufficient quality to be published in Nature Communications.